# Experimental warming causes mismatches in alpine plant-microbe-fauna phenology

Rui Yin[1,10], Wenkuan Qin[1,10], Xudong Wang[1], Dong Xie[2], Hao Wang [3], Hongyang Zhao[1], Zhenhua Zhang [4], Jin-Sheng He [1,2], Martin Schädler [5,6], Paul Kardol [7,8], Nico Eisenhauer [6,9] & Biao Zhu [1] ✉

Long-term observations have shown that many plants and aboveground animals have changed their phenology patterns due to warmer temperatures over the past decades. However, empirical evidence for phenological shifts in alpine organisms, particularly belowground organisms, is scarce. Here, we investigate how the activities and phenology of plants, soil microbes, and soil fauna will respond to warming in an alpine meadow on the Tibetan Plateau, and whether their potential phenological changes will be synchronized. We experimentally simulate an increase in soil temperature by 2–4 °C according to future projections for this region. We find that warming promotes plant growth, soil microbial respiration, and soil fauna feeding by 8%, 57%, and 20%, respectively, but causes dissimilar changes in their phenology during the growing season. Specifically, warming advances soil faunal feeding activity in spring and delays it in autumn, while their peak activity does not change; whereas warming increases the peak activity of plant growth and soil microbial respiration but with only minor shifts in their phenology. Such phenological asynchrony in alpine organisms may alter ecosystem functioning and stability.

Global warming may not only change the biodiversity and community structure of both above- and below-ground organisms, but also their activity and phenology, including changes in the timing and magnitude of activity peaks[1–4]. Phenological changes in biological activities, in turn, may further have feedback effects on the climate[5,6]. Warming has been reported to have particularly strong effects on ecosystems at high elevations and altitudes[7,8], where ecological processes are rather sensitive to environmental changes and strongly limited by cold temperatures[9,10]. Warming, however, can alleviate the thermal constraints[11,12], thus biological activities are expected to increase under predicted future warming scenarios. Further, the effects of warming on these biological activities may depend largely on soil moisture[13]. Water scarcity during the dry summer months could be a limiting factor that offsets or diminishes the increases in biological activities caused by warming[14].

Since temperature is one of the main drivers of the phenological cycle of organisms, on-going climate change may alter biological phenology with consequences for ecosystem functioning[2,4,15–17]. In

[1]Institute of Ecology, College of Urban and Environmental Sciences, and Key Laboratory for Earth Surface Processes of the Ministry of Education, Peking University, Beijing, China. [2]State Key Laboratory of Herbage Improvement and Grassland Agro-Ecosystems, and College of Pastoral Agriculture Science and Technology, Lanzhou University, Lanzhou, China. [3]State Key Laboratory of Herbage Improvement and Grassland Agro-Ecosystems, and College of Ecology, Lanzhou University, Lanzhou, China. [4]Qinghai Haibei National Field Research Station of Alpine Grassland Ecosystem, and Key Laboratory of Adaptation and Evolution of Plateau Biota, Northwest Institute of Plateau Biology, Chinese Academy of Sciences, Xining, China. [5]Helmholtz Centre for Environmental Research—UFZ, Department of Community Ecology, Theodor-Lieder-Strasse 4, 06110 Halle (Saale), Germany. [6]German Centre for Integrative Biodiversity Research (iDiv) Halle-Jena-Leipzig, Puschstr. 4, 04103 Leipzig, Germany. [7]Department of Forest Mycology and Plant Pathology, Swedish University of Agricultural Sciences, 750-07 Uppsala, Sweden. [8]Department of Forest Ecology and Management, Swedish University of Agricultural Sciences, 901-83 Umeå, Sweden. [9]Institute of Biology, Leipzig University, Puschstr. 4, 04103 Leipzig, Germany. [10]These authors contributed equally: Rui Yin, Wenkuan Qin. ✉e-mail: biaozhu@pku.edu.cn

contrast to extensive recent work on the phenology of plants and aboveground animals in the context of climate warming[16,18–24], soil biota have rarely been included in phenological studies, despite their high sensitivity to environmental changes[1]. In cold biomes, for example, long-term records have shown spring advancement and/or autumn postponement in plant growth in response to warming[10,12,21,25]. The resulting longer plant growing season may increase gross primary productivity and net $CO_2$ uptake[26], associated with a negative feedback to climate change[6]. However, whether similar phenological shifts may also occur in soil biota, and whether the phenology of different groups of organisms (such as plants, soil microbes, and soil fauna) would respond in synchrony to warming in a given region or ecosystem, remains largely unknown. Therefore, warming experiments on above-belowground communities could provide novel insights into this topic.

Here, we used a warming experiment in an alpine meadow of the Tibetan Plateau (Fig. 1), where we manipulated soil temperature according to projections for the year 2100 (ambient temperature +2–4 °C)[27] using whole-soil-profile heating rods. Based on PhenoCam-derived Normalized Difference Vegetation Index (NDVI) and in-situ investigations (combined with the dynamic chamber method and bait-lamina method), the seasonal dynamics of plant growth (represented by NDVI data), soil microbial respiration (represented by root-free soil respiration), and soil fauna feeding (represented by the feeding of soil invertebrate detritivores, e.g., earthworms, isopods, millipedes, enchytraeids, Collembola, and oribatid mites) were investigated to explore phenology shifts in the context of continuous warming. We tested the following hypotheses: H1: Experimental warming increases biological activities (i.e., plant growth, soil microbial respiration, and soil fauna feeding) during the growing season, due to warming-induced releases of low-temperature constraints[11,12]. H2: The effects of

experimental warming on biological activities would vary over the growing season. More specifically, the increases in biological activities would be particularly pronounced in the early and late-growing seasons, resulting in earlier spring phenology and delayed autumn phenology. Such a phenological shift pattern of alpine plant growth in response to natural climate warming has been revealed over the past few decades[10].

## Results and discussion

We found that warming significantly increased the activities of plant growth, soil microbial respiration, and soil invertebrate detritivore feeding by 8%, 57%, and 20%, respectively (Fig. 2a–c and Table 1). However, warming may not always increase biological activities in cold biomes, particularly in dry habitats or under dry conditions[12,14]. By contrast, for the alpine meadow we studied, the growing season is characterized by wet and cool climatic conditions. During the growing season, the heated plots achieved a warming of about 2.2 °C at 5 cm soil depth (Supplementary Fig. 1a). However, warming slightly reduced water content in the surface soil (0–10 cm) by about 6% across all plots during the whole growing season, with only a significant reduction in the late-growing season (September and October) (Supplementary Fig. 1b). Ectothermic organisms, such as plants and soil biota, generally exhibit greater activity with warming under sufficient levels of soil water content due to their increased metabolic demands[28,29]. Interestingly, the increased activity of plant growth in the context of warming did not represent a significant increase in their total biomass during the alpine growing season (Supplementary Fig. 2a). This result is in line with a former study in this region, suggesting that climate warming increases plant growth activity but shortens plant growth period, resulting in unchanged plant biomass[10]. Likewise, recent studies in temperate deciduous forests have confirmed that, in general,

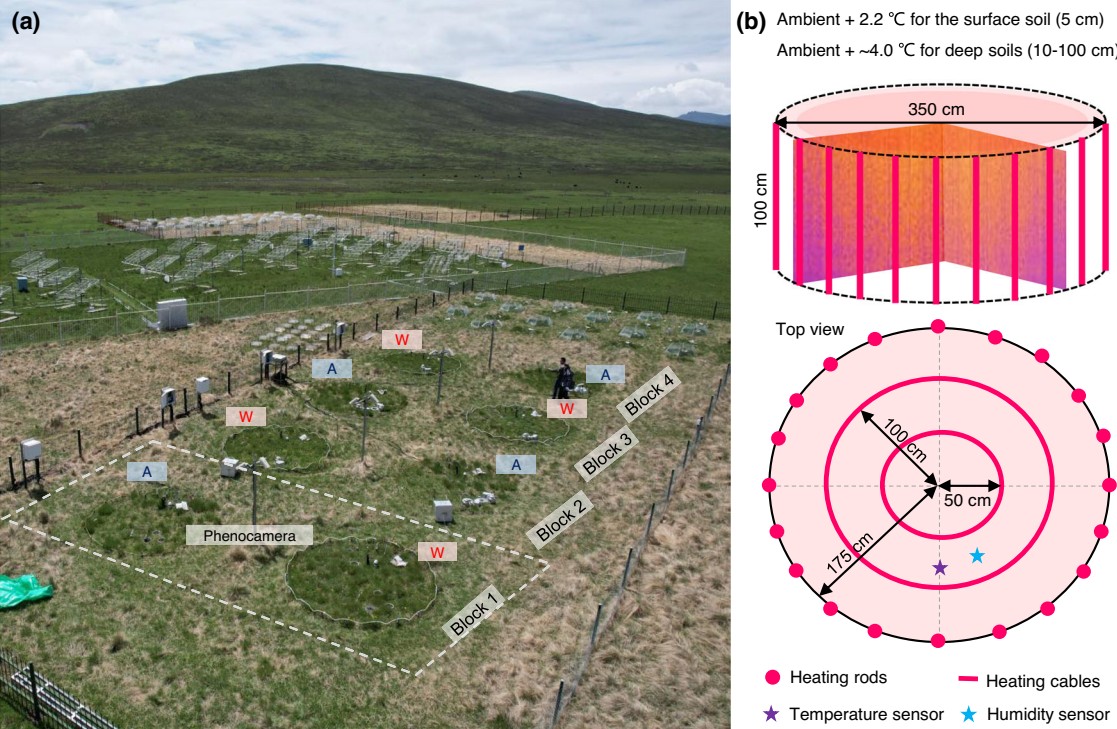

**Fig. 1 | Experimental set-up. a** Aerial view of the whole-soil-profile warming experiment conducted at the Haibei Alpine Grassland Ecosystem Research Station on the Qinghai-Tibetan Plateau, China. The experiment involved two climate treatments, namely warming vs. ambient temperature, arranged into eight plots, with four replicates for each treatment. Each plot was equipped with a PhenoCamera. **b** A three-dimensional layout diagram depicting the arrangement of 20

heating rods, each 100 cm deep, around a plot with a diameter of 3.5 m, along with two circular heating cables situated 5 cm below the soil surface at radii of 0.5 and 1 m from the plot center. This method resulted in an average warming effect of -2.2 °C for the surface soil (5 cm) and 4.0 °C for deeper layers of the soil profile (10–100 cm). Temperature and moisture sensors were used to monitor soil temperature and water content, respectively (Supplementary Fig. 1).

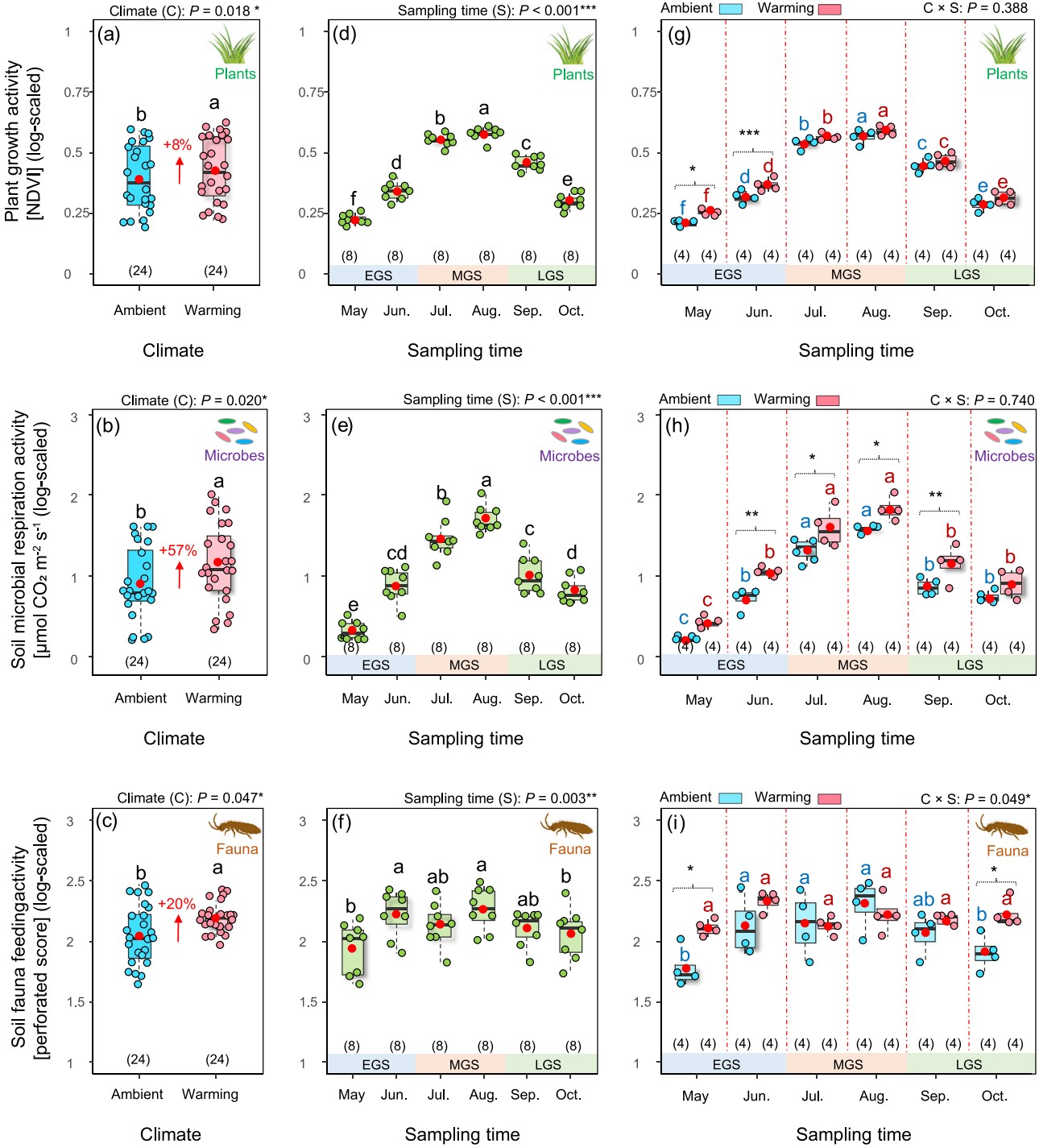

**Fig. 2 | Climate and temporal effects on activities of plants, soil microbes, and soil fauna.** Effects of climate (ambient vs. experimental warming) on (**a**) plant growth activity ($n = 24$), (**b**) soil microbial respiration activity ($n = 24$), and (**c**) soil fauna feeding activity ($n = 24$). Effects of sampling time (i.e., 6 months of the growing season) on (**d**) plant growth activity ($n = 8$), (**e**) soil microbial respiration activity ($n = 8$), and (**f**) soil fauna feeding activity ($n = 8$). Interactive effects of climate and sampling time on (**g**) plant growth activity ($n = 4$), (**h**) soil microbial respiration activity ($n = 4$), and (**i**) soil fauna feeding activity ($n = 4$). Data were analyzed using linear mixed effects models (see "Methods" for details). Based on two-sided tests for multiple comparisons by FDR corrections, different lowercase letters at the top of the boxes indicate significant differences among respective groups ($p < 0.05$) (**a**–**i**); *$p < 0.05$, **$p < 0.01$, and ***$p < 0.001$ indicate significant differences between ambient and warming climates in specific sampling time (**g**–**i**). Box center lines represent the median, box limits represent the upper and lower quartiles, whiskers represent the $1.5 \times$ interquartile range from the 25th and 75th percentiles, red dots represent the mean, and jittered points represent biologically independent samples for each group. EGS early growing season (spring = May–June), MGS middle growing season (summer = July–August.), LGS late-growing season (autumn = September–October).

the gross primary production is likely to remain stable, although warming could induce shifts in the timing (e.g., earlier phenology) and increases in the maximum values of plant growth[21,30]. Warming may have led to increased water stress during the middle and/or late alpine

growing season[31]. Consistent with this mechanism, we found a significant reduction in soil water content during the late-growing season (i.e., in September and October, Supplementary Fig. 1b), which could be attributed to (1) the reduction induced by warming per se, and (2)

**Table 1 | Results (*F*-values) of linear mixed effects models (Type III ANOVA with Kenward–Roger's method) testing the effects of climate (ambient vs. experimental warming), sampling time (i.e., 6 months of the growing season), and their interaction on plant growth activity, soil microbial respiration activity, and soil fauna feeding activity (all log-scaled)**

| Treatment (Numerator and denominator *df*) | Plant growth activity | | Soil microbial respiration activity | | Soil fauna feeding activity | |
|---|---|---|---|---|---|---|
| | *F*-value | *p* value | *F*-value | *p* value | *F*-value | *p* value |
| Climate (1,3) | **22.16*** | 0.018 | **20.47*** | 0.020 | **10.56*** | 0.047 |
| Sampling time (5,30) | **909.06*** | <0.001 | **135.27*** | <0.001 | **4.61**** | 0.003 |
| Climate × Sampling time (5,30) | 1.09 | 0.388 | 0.55 | 0.740 | **2.54*** | 0.049 |
| Contrasts for sampling time | *z*-value | *p* value | *z*-value | *p* value | *z*-value | *p* value |
| EGS vs. MGS | **−56.94*** | <0.001 | **−24.38*** | <0.001 | −1.99 | 0.114 |
| EGS vs. LGS | **−18.09*** | <0.001 | **−10.55*** | <0.001 | −0.10 | 0.995 |
| MGS vs. LGS | **38.86*** | <0.001 | **13.84*** | <0.001 | 1.90 | 0.140 |

Results (*z*-values) of orthonormal contrasts "EGS vs. MGS", "EGS vs. LGS", and "MGS vs. LGS" showing the differences for each activity among three different stages of the growing season.
*EGS* early growing season (spring = May and June), *MGS* middle growing season (summer = July and August), *LGS* late-growing season (autumn = September and October).
The statistical tests are two-sided, and significant effects with $*p < 0.05$; $**p < 0.01$; $***p < 0.001$ are indicated in bold font.

greater consumption of available soil water by the warming-induced higher spring biomass. By contrast, the increased activities in soil biota were accompanied by an increase of microbial and faunal biomass under warming (Supplementary Fig. 2b, c). As a result, the increased soil biological activities may trigger a positive feedback to climate change through enhancing carbon loss from soil[32,33].

The activities of plants and soil biota showed strong temporal dynamics (Fig. 2d–f and Table 1). Uniformly, their activities were highest during the warmest middle growing season (orthonormal contrasts, Table 1). However, warming significantly enhanced plant growth activity in the early growing season (June and July) (Fig. 2g), and consistently increased soil microbial respiration activity from late spring (June) to early autumn (September) (Fig. 2h). By contrast, the responses of soil fauna feeding activity to warming varied across the seasons, with significant increases in their activity only in early spring (May) and late autumn (October) (Fig. 2i). In general, soil fauna are more sensitive to temperature changes than soil microbes[34]. Within the thermal limits of soil fauna, warming may enhance their feeding activity more profoundly in cooler months; however, warming-induced higher activity in their feeding might be offset by droughts in warmer months[14].

Importantly, we found that the responses of these alpine organisms to warming were asynchronous (Fig. 3). Specifically, experimental warming caused an advanced spring phenology and a delayed autumn phenology for soil fauna without any significant change in the peak activity; whereas the peak activity of plants and soil microbes increased in the middle of the growing season without any significant phenological shifts (Fig. 3a–c). Notably, climate warming has indeed advanced spring phenology of plant growth globally by an average of 4–6 days over the past few decades, but the phenological shifts might be less pronounced as climate change continues[35,36]. Further, we speculate that advanced spring and delayed autumn phenology of soil faunal activity could increase their vulnerability to abrupt temperature changes. Namely, soil fauna could be more vulnerable to freezing in early spring and late autumn, as has been already demonstrated for plants with earlier spring phenology[25].

These contrasting phenological responses among different groups of organisms indicate warming-induced phenological mismatches. These mismatches may result in disruptions of trophic and non-trophic interactions of coexisting organisms through changes in their food web structure and energy fluxes, thereby degrading ecosystem functioning and stability[37]. In general, we found a correlation between the activities of the three organisms (Fig. 4a–c). For example, soil microbial respiration was positively correlated with plant growth across all plots and dates; however, these correlations were significant under ambient climate but not under warming climate (Fig. 4a). The disappearance of such a relationship under warming

can be explained by the phenological mismatches-induced disruption of energy and nutrient fluxes, as well as trophic interactions of organisms[37–39].

Furthermore, phenophase temperature responses were reported to depend largely on soil water availability, with greater temperature sensitivity under wetter conditions[40]. Therefore, we additionally tested whether the effects of warming on biological activities could be explained by the changes in soil microenvironment. Pearson correlation tests were performed, including data on monthly mean temperature and water content of surface soil (0–10 cm) across the plots. These analyses showed that the activities of soil fauna and particularly plants and soil microbes were all positively correlated with soil temperature but not with soil water content (Supplementary Fig. 4a–f). This result might be largely due to the fact that our warming treatment significantly increased soil temperature but only negligibly changed soil water content (Supplementary Fig. 1).

To our knowledge, the findings of this study provide the first evidence of phenological mismatches between above- and below-ground alpine organisms due to climate warming. Notably, these significant warming effects were already observed 3 years after the establishment of the experimental platform, indicating rather fast but dissimilar responses of different organisms to warming. Given the potential significance of warming-induced positive feedback effects on[41] as well as the high vulnerability of higher latitude and altitude environments to climate change[42], long-term observations are needed to study if such phenological mismatches are a transient or stable consequence of a changing climate. Previous work has shown that adaptation processes may mitigate the responses of plants and soil microbes to warming[43,44], while long-term studies in soil animal physiology and phenology are lacking. In light of these results, warming-induced phenological mismatches among organisms may have far-reaching implications for trophic interactions, food web dynamics, and energy fluxes, resulting in ecosystem-level consequences for ecosystem carbon cycling and feedbacks to climate change[45,46]. Quantifying these phenology-related implications represents a critical future research challenge, and this emerging research field deserves greater attention to better predict the structural and functional ecosystem shifts in a changing climate.

## Methods
### Site description and experimental design
We conducted our study at the Haibei Alpine Grassland Ecosystem Research Station, utilizing a whole-soil-profile warming experiment in an alpine meadow on the Qinghai-Tibetan Plateau, China (37°29′–37°45′ N, 101°12′–101°23′ E, 3200 m a.s.l.). The climate of this study area is a continental monsoon characterized by short, cool

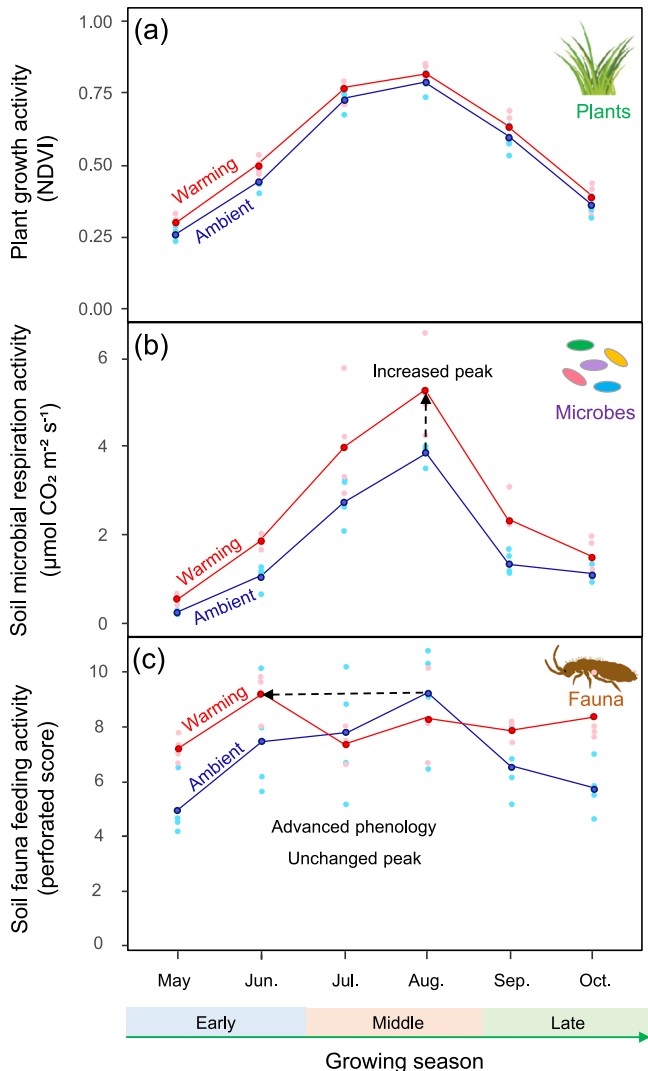

**Fig. 3 | Phenology of activities of plants, soil microbes, and soil fauna during the growing season under ambient vs. warmed climate conditions.** Warming increased the peak values of plant growth and soil microbial respiration activities in the middle of the growing season without much shifts in their phenology (**a**, **b**). Warming advanced spring phenology and delayed autumn phenology of soil fauna feeding activity, with an unchanged activity peak during the growing season (**c**). The monthly average activities of these organisms were connected by straight lines (with the corresponding colored sampling points, $n = 4$) for both ambient and warming climates. EGS early growing season (spring = May–June), MGS middle growing season (summer = July–August.), LGS late-growing season (autumn = September–October).

summers and long, cold winters, with a 6-month growing season (from mid-April to mid-October). The average annual air temperature is about 1.1 °C, and the highest temperature occurs in July or August, and the lowest temperature occurs in January. Average annual precipitation over the last three decades was about 485 mm, with most annual precipitation (84%) falling from May to September[10]. The alpine meadow is dominated by grasses, such as *Stipa aliena*, *Elymus nutans*, and *Helictotrichon tibeticum*, mixed with forbs, including *Gentiana straminea*, *Tibetia himalaica*, *Saussurea pulchra*, and *Medicago ruthenica*, and sedges, such as *Kobresia humilis* and *Carex przewalskii*[47]. The soil type is Mat–Cryic Cambisol, with soil organic carbon (SOC): $72.9 \pm 17.6$ g kg$^{-1}$, extractable organic carbon (EOC): $433.6 \pm 38.3$ mg kg$^{-1}$, total nitrogen (TN): $7.2 \pm 1.6$ g kg$^{-1}$, extractable total nitrogen (ETN): $62.7 \pm 6.8$ mg kg$^{-1}$, inorganic nitrogen (IN): $7.8 \pm 0.9$ mg kg$^{-1}$, ammonium nitrogen (NH$_4^+$-N): $4.2 \pm 0.5$ mg kg$^{-1}$,

nitrate nitrogen (NO$_3^-$-N): $3.6 \pm 0.8$ mg kg$^{-1}$, and pH value: $7.6 \pm 0.1$ in the surface soil (0–10 cm) of experimental plots.

The experiment consists of four blocks. Each block contains two circular plots (with a diameter of 3.5 m), one of which is subjected to warming climate, while the other one is under ambient climate (Control); that is, 4 replicated plots for each climate treatment (see Fig. 1a for more details of the experimental set-up). The reason why we chose to group the plots in pairs into spatial blocks was that (1) there are natural gradients (e.g., microtopography, plant community composition) in the field site; (2) the paired warmed and control plots were both steered by a single data logging system coupled to a warming system, which could cause data dependency. For each warmed plot, 20 heating rods were placed into the soil to a depth of 100 cm, with two additional circular heating cables 5 cm below the soil surface at radii of 0.5 and 1 m from the plot center (Fig. 1b), to achieve an observed temperature increase of about 2.2 °C for the surface soil (5 cm) and about 4.0 °C for deep layers of the soil profile (10–100 cm). For the control plots, the same installations were set up to avoid potential side effects of the installations in the warmed plots.

Although warming may inevitably affect precipitation patterns, we did not manipulate the precipitation in this study due to inconsistent precipitation projections for this region. The experimental plots were established in 2017, and the warming treatments began on June 18, 2018. We conducted this study in the growing season of 2021 (from May to October), i.e., 3 years after the beginning of the experiment. We measured soil temperature and soil water content by a custom-made thermistor and a PR2/6 sensor (Delta-T Devices Ltd., UK) at 10-min intervals, respectively. The corresponding data are shown in Supplementary Fig. 1.

## Monitoring of biological activities

We investigated the seasonal dynamics of plant growth activity in each plot based on PhenoCam-derived Normalized Difference Vegetation Index NDVI data (Supplementary Fig. 3) in combination with field-measured plant biomass (described in the following section), to explore shifts in the timing of plant growth activity in response to climate warming[2,10]. Specifically, we adopted the technique of repeated digital photographs for plant growth activity and its phenology monitoring (using NetCam SC phenocamera, StarDot Technology, Buena Park, CA, USA, Fig. 1a). PhenoCam-derived NDVI obtained from NIR-enabled digital camera represents a valuable complement of NIR channel to the traditional greenness index or green chromatic coordinate, derived from visible 3-channel (RGB: red, green, blue) imaging[48]. The method for the extraction of PhenoCam-derived NDVI from a stack of images including the following steps: (1) the PhenoCam (NetCam SC; StarDot Technologies, Buena Park, California, USA) obtained one pair of RGB and RGB + NIR images per shot[48]. Digital Number (DN) values ($R_{DN}$, $G_{DN}$, $B_{DN}$) of red, green and blue channels were extracted from RGB images; DN values [(RGB + NIR)$_{DN}$] of red, green, blue, and near infrared channels were extracted from RGB + NIR images. (2) The DN values extracted from RGB and RGB + NIR images were respectively calibrated to $R_{DN\_E}$, $G_{DN\_E}$, $B_{DN\_E}$, and (RGB + NIR)$_{DN\_E}$ according to the exposure value[49]. (3) After calculating the calibration value of NIR$_{DN\_E}$, phenological camera NDVI$_C$ was calculated[50]: NDVI$_C = \frac{NIR_{DN\_E} - R_{DN\_E}}{NIR_{DN\_E} + R_{DN\_E}}$. (4) In order to be comparable with field spectral sensors or satellite derived NDVI, we scaled the digital number-derived NDVI$_C$ to normal NDVI values using a linear regression ($y = 0.9445x + 0.0471, n = 18$)[51]. We established this linear regression based on the data from simultaneous measurements of spectral reflectance-derived NDVI$_R$ (measured by Phenological camera, StarDot Technologies, Buena Park, California, USA) and digital number-derived NDVI$_C$ during the growing seasons of 2018 and 2019 (Supplementary Fig. 5) for the same regions of interest in our studied alpine grassland.

In order to establish high quality NDVI time series, we conducted preprocessing procedures for the obtained normal NDVI values. First,

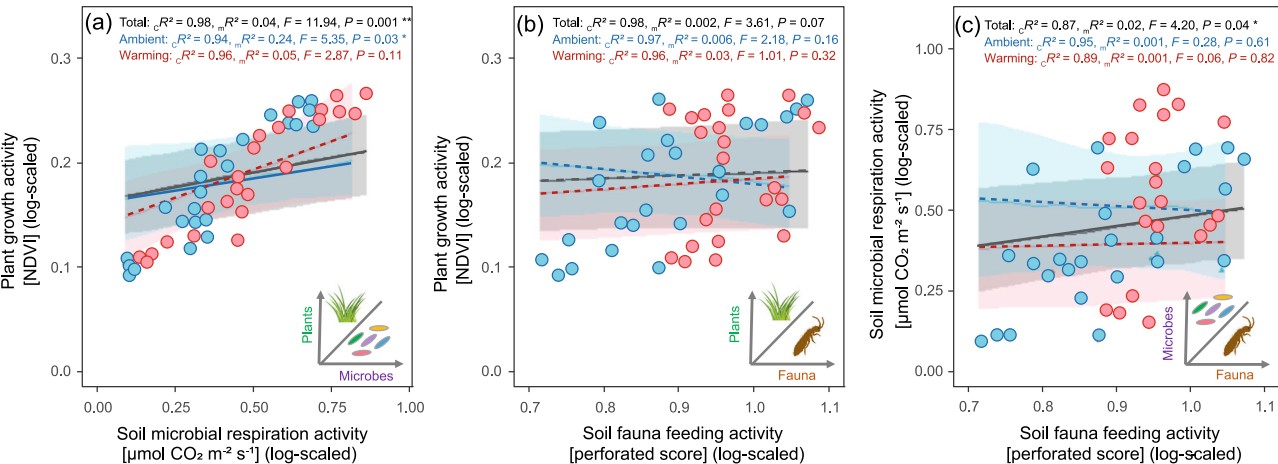

**Fig. 4 | Relationships among activities of plants, soil microbes, and soil fauna.** Correlations ($R^2$) between plant growth activity and soil microbial respiration activity (**a**), between plant growth activity and soil fauna feeding activity (**b**), between soil microbial respiration activity and soil fauna feeding activity (**c**). The presented statistics are conditional $R$ squares ($_cR^2$), marginal $R$ squares ($_mR^2$), $F$-values, and $p$ values from linear mixed effects models. The statistical tests are two-sided, and significant effects are indicated by $^*p < 0.05$, $^{**}p < 0.01$. Points represent biologically independent samples (total points, $n = 48$; blue points for ambient climate, $n = 24$; red points for warming climate, $n = 24$). Lines and error bands depict the best-fit trendline and the 95% confidence interval of the linear regression, respectively. Blue lines represent ambient climate, red lines represent warming climate, and gray lines are for all data points together.

we deleted 0- and negative values (NDVI ≤ 0, usually due to snow cover in winter or light deficiency-derived obvious outliers, like in cloudy days and early morning[52]). Based on the daily dynamics of NDVI (Supplementary Fig. 6), we then selected the daily maximum NDVI values from the remaining values (NDVI > 0) between 11:30 and 14:30 for each day to calculate a mean daily NDVI. Their 95 and 5% quantiles were treated as the upper and lower limits of NDVI sequences to constrain mean daily NDVI values in a reasonable range[51]. We finally used the Savitzky–Golay filter in the R "phenofit" package to filter and denoise NDVI sequences[53]. We used a double logistic (DL) function to obtain the smooth seasonal dynamic curves of NDVI, and to interpolate the missing values[54]. For better understanding, the comparison for the raw, filtered, and fitted NDVI sequences under ambient and warming plots is shown in Supplementary Fig. 7, and a brief flowchart of NDVI data processing is additionally provided in Supplementary Table 1. In this study, we used the diurnal scale NDVI data to calculate the monthly mean of plant growth activity for the growing season months (i.e., from May to October) of 2021.

The soil is a clay-loam, and its average thickness is about 100 cm (with very little SOC below 70 cm); we therefore measured soil microbial respiration ($\mu mol\ CO_2\ m^{-2}\ s^{-1}$) to a depth of 65 cm in line with previous studies[55–57]. Specifically, we inserted a PVC collar (70 cm long, 10 cm in diameter) into the bulk soil (65 cm depth) of each plot in June 2016. This method has been demonstrated to be efficient in cutting off old plant roots and prevented new roots from growing inside the tubes[58,59]. We used a SF-9000 soil $CO_2$ efflux analyzer (LICA United Technology Ltd., China) attached to a chamber to measure soil $CO_2$ efflux twice (for the mean) on a sunny day from 8:00 to 11:00 at the beginning of each month of 2021 growing season (i.e., from May to October).

We measured soil fauna activity to a depth of 10 cm using standard bait-lamina strips[60]. Standard bait-lamina strips (1 mm thick × 6 mm wide × 120 mm long) with 16 holes (Ø 1.5 mm, at 5 mm intervals) were made from PVC rigid plastic (Terra Protect, Berlin). We filled the holes with a bait substrate consisting of 70% cellulose powder, 27% wheat bran, and 3% activated carbon. The bait substrate is primarily consumed by earthworms, isopods, millipedes, enchytraeids, Collembola, and oribatid mites; whereas microbes contribute very little to the perforation and loss of bait substrate in a short period of study[14,34].

To insert the strips into the soil without dislodging the bait in the holes, or breaking the strips, we used a steel knife to create a slit 10 cm

deep, with the same width and thickness as the bait-lamina strips. We inserted six strips into the middle of each plot at a distance of 10 cm from one another to account for some potential spatial heterogeneity. We placed strips in the middle of each month during the growing season of 2021 (i.e., from April to October). After 30 days of exposure, we gently removed all bait-lamina strips from the soil (and subsequently replaced with new bait strips), and immediately assessed the perforation in the field. Specifically, we carefully inspected all holes (combined for all six strips used per plot) and scored them as 0 (no feeding activity), 0.5 (intermediate feeding activity), or 1 (high feeding activity)[14,34]. The score of bait consumption can therefore range from 0 (no feeding activity) to 16 (maximum feeding activity) per strip. We calculated the mean bait consumption of the six strips per plot prior to statistical analysis.

### Measurements of other potential explanatory variables
**Plant biomass.** Mid-August 2021, we randomly chose two 50 cm × 50 cm quadrats in each plot to investigate aboveground productivity, and clipped all plants of the two diagonal 25 cm × 25 cm sub-quadrats in each quadrat to evaluate shoot biomass (g m$^{-2}$). As 96% of root biomass is typically found in the upper 30 cm[61], we took five soil cores (Ø 5 cm, 30 cm depth) in each plot to evaluate root biomass (g m$^{-2}$). Total plant biomass (g m$^{-2}$) is equal to the sum of shoot biomass (g m$^{-2}$) and root biomass (g m$^{-2}$).

**Soil microbial biomass carbon.** Mid-August 2021, we randomly collected five soil samples (Ø 3 cm, 10 cm depth) from each plot and evenly mixed them (as a composite sample for this plot) to determine the content of microbial biomass carbon by the chloroform fumigation-extraction method[62]. Briefly, 4 g fresh soil was fumigated for 48 h with ethanol-free $CHCl_3$ to kill all soil microbes. After that, both fumigated and unfumigated soil samples were extracted using 0.5 mol l$^{-1}$ $K_2SO_4$ (soil:solution = 1:4), and the extracting solution was then filtered through a 0.45 mm filter membrane. The extractable organic carbon in the filtered solution was analyzed by a TOC/TN analyzer (Multi N/C 3100, Analytik Jena, Germany). We calculated soil MBC as the difference in extractable C concentrations between fumigated and unfumigated samples using a conversion factor of 0.45.

**Soil fauna.** Mid-August 2021, we collected two soil cores (Ø 6 cm, 10 cm depth) from each plot and placed them together (as a composite

sample for this plot) to extract soil fauna using a Kempson method[63]. Note that this method is limited to the extraction of (micro)arthropods, but other taxa (such as nematodes, earthworms and macroarthropods) are also important for ecological processes such as decomposition. We used a VHX-Digital microscope to classify, count and measure the body length of all observed individuals that were dominated by Collembola and Acari. We calculated individual mean biomass ($M$, μg) using the formula Log $M = a + b \times$ Log $L$, with $L$ as the individual mean body length (μm), with Collembola: $a = 1.34$, $b = 1.99$; Oribatida: $a = 2.12$, $b = 2.71$; Prostigmata: $a = 2.12$, $b = 2.8$[64]. Total biomass is equal to the number of individuals (abundance) multiplied by the individual mean biomass.

## Statistical analyses

We used linear mixed effects models (LMMs, Type III ANOVA with Kenward–Roger's method) to analyze the fixed effects of climate (ambient vs. warming), sampling time (May, June, July, August, September, October), and their interactions on biological activities (i.e., plant growth, soil microbial respiration, soil fauna feeding) and other potential explanatory variables (i.e., soil fauna abundance and biomass, microbial biomass carbon, plant biomass) using the *lmer* () function in R "lme4" package[65]. In the models, plot nested within block served as a random effect. We performed the Shapiro-Wilk test and Levene's test to test the normality of the model residuals and the homogeneity of variance, respectively. If not normally distributed, the data were log-transformed log($x + 1$) and the analyses were redone. Further, we conducted the orthonormal contrasts to test for the differences in biological activities among three different stages of the growing season (i.e., early-, middle-, and late-growing seasons) using the *glht* () function in R "multcomp" package. In the models, we performed FDR (false discovery rate) corrections for multiple comparisons of means to reveal significant differences ($p < 0.05$) among respective groups. Additionally, we used the same linear mixed models to test for the differences in soil temperature and soil water content between ambient and warmed plots for each month. Finally, we built the same linear mixed model with the same random effects structure to test the correlations (1) among biological activities, and (2) between each biological activity and abiotic factors (i.e., daily mean temperature and water content). We performed all statistical analyses using R version 4.2.2[66] with R studio interface.

## Reporting summary

Further information on research design is available in the Nature Portfolio Reporting Summary linked to this article.

## Data availability

The data that support the findings of this study are available under CC-BY 4.0 license from Figshare. Data on the biological activities (i.e., plant growth, soil microbial respiration, soil fauna feeding), and soil abiotic conditions (i.e., soil temperature, soil moisture) are available from Figshare: https://doi.org/10.6084/m9.figshare.22357921.v2; data on the other explanatory variables (i.e., plant biomass, soil microbial biomass carbon, soil fauna biomass) are available from Figshare: https://doi.org/10.6084/m9.figshare.22358158.v1; Data on the initial soil chemical properties (i.e., soil organic carbon, extractable organic carbon, total nitrogen, extractable total nitrogen, inorganic nitrogen, ammonium nitrogen, nitrate nitrogen, pH value) are available from Figshare: https://doi.org/10.6084/m9.figshare.22358254.v1.

## Code availability

Programming code for the linear mixed effects model, orthonormal contrasts test, and correlation analysis is available under CC-BY 4.0 from Figshare: https://doi.org/10.6084/m9.figshare.22424107.v1.

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

## Acknowledgements

This study was supported by the National Natural Science Foundation of China (31988102, 32101375, 42141006) and the China Postdoctoral Science Foundation (2021M700231). N.E. acknowledges support of iDiv funded by the German Research Foundation (DFG–FZT 118, 202548816)

and funding by the DFG (Ei 862/29-1 and Ei 862/31-1), the European Research Council (ERC) under the European Union's Horizon 2020 research and innovation program (grant agreement no. 677232), and by the Saxon State Ministry for Science, Culture and Tourism (SMWK), Germany—[3-7304/35/6-2021/48880]. P.K. acknowledges funding from the European Research Council (ERC) under the European Union's Horizon 2020 research and innovation program (grant agreement no. 864287—THRESHOLD—ERC-2019-COG). We thank Yawen Shen, Qiufang Zhang, and Ying Chen for their constructive feedback, and the staff at the Haibei station of Chinese Academy of Sciences for providing logistical support.

## Author contributions

B.Z. and R.Y. conceived the study. W.Q., H.Z. and X.W. conducted the field work of measuring soil microbial and faunal activities. H.W., D.X. and J.S.H. provided the plot-level NDVI data from plant phenology cameras, and the guidance on NDVI data processing. R.Y. conducted the lab work, analyzed the data, and wrote an initial draft of the manuscript with P.K., N.E., M.S. and B.Z. All authors contributed to the further revising and editing of the manuscript.

## Competing interests

The authors declare no competing interests.
