## [Peer Review File · Nature Communications]

Response from Reviewers

Reviewer #1 (Remarks to the Author):

The authors used a soil warming experiment in an alpine system to assess how different groups of organisms (plants, soil microbes, and soil fauna) respond to warming not only in terms of their activity and biomass but also the timing of this activity. The authors find that while plants and soil microbes show an increased peak of activity under warming, this happens at the same time as in the control plots, namely in the middle of the summer season. Soil fauna, in contrast show a shift in the timing of the activity peak from mid season to early and late season, without the two new peaks differing in magnitude from that in the control plots. They conclude that this phenological mismatch under warming could have important consequences for ecosystem functioning in such alpine systems. I agree with the authors that this is an important topic and experiments such as the one presented here are the way forward to increase our limited understanding of the effects of climate warming on soil organisms and processes. In general, I like the straightforward setup of the experiment and the uncomplicated way in which the results are presented. In general the manuscript is well prepared. However, I have a few major and some minor comments that preclude me to recommend the manuscript in its current form for publication in Nature Communications. If the authors can address these concerns, I think the manuscript can represent an important contribution to the development of this understudied field.

Major comments:

- Would the fact that you measured root biomass to a depth of 30 cm, soil microbial activity from a depth of 65 cm and soil fauna activity only to a depth of 10 cm have influenced the results, and if so, how?
- The soil fauna activity was estimated with bait-lamina strips and their abundance with Kempson extractions. Both methods are quite selective, mainly targeting soil (micro)arthropods, but ignoring abundant groups such as nematodes, rotifers, tardigrades etc., many of which feed on microbes and therefore would be relevant to this study. This fact should at least be highlighted and its possible implications discussed.
- Whereas the linear mixed models seem adequate, the correlation analyses do not. They seem to have been performed on data from across plots and months, essentially violating the assumption of data independence. Furthermore, all data for the soil organisms groups were log-scaled, presumably to improve normality of model residuals, but this is not explained nor are results of normality tests presented. See detailed comments about statistics below.

Detailed comments:

- Line 24: Suggest rephrasing as "many plants and aboveground animals" as otherwise the reader may think "aboveground" also applies to "plants", which are both above- and belowground.
- Line 33: Suggest removing "the activities"
- Line 35: Suggest rephrasing as "warming advanced soil fauna activity in the spring and delayed it in the autumn"
- Line 39. The word "mismatch" is perhaps not ideal here, as from the previous sentence I gather that peak activity still occurs at the same time for all three tested groups of organisms.
- Line 46: Suggest replacing "altitudes" with "elevations"
- Line 48: "biological activities are expected to increase with phenological shifts" is slightly ambiguous. Do you mean that as phenological shifts increase, biological activities are expected to increase in tandem, or just that biological activities are expected to increase anyhow, accompanied by phenological shifts?
- Line 51: Why soil moisture and not moisture in general? This paragraph is still talking about both above- and belowground organisms.
- Line. 54: This is a bold claim about "organisms" in general with a single reference to a paper about plant phenology in particular. I can come up with plenty of examples of phenology driven by day length and not temperature. Please tone down or be more precise.
- Line 56. Same as previous comment, I think that "aboveground plants" makes no sense as plants are simultaneously above- and belowground (bar some epiphytes and other exceptions).
- Line 63: "(i.e., plants, soil microbes, and soil fauna)" reads strange, as if this were the only way

one could describe different groups of soil organisms. Perhaps "such as plants, soil microbes, and soil fauna" would be better.

- Line 70: Suggest changing "whole-soil profile" to "whole-soil-profile"

- Line 71: "(i.e., NDVI)" suggests remote-sensing-based technology is equal to NDVI. NDVI is an index of plant productivity based on relationships among visible red and near infrared light reflected off of surfaces. Suggest changing to something like "and remote sensing measurements (e.g., NDVI)"

- Line 77: This sentence has 52 words, no less. I suggest splitting into smaller, more digestible sentences.

- Line 85: "soil fauna feeding" is too vague at this point without having read the methodology. Add briefly what you actually measured here.

- Line 101: What is meant by "increased peaks"? Increased growth speed over a certain time interval?

- Line 115: "off set" should be "offset"

- Line 119: The information that Fig. 2 is based on smoothing of the monthly mean values for ambient and warmed conditions is a bit limited. What was the smoothing procedure?

- Line 122: Should the reference to Supp. Fig. 4 here have been to Supp. Fig. 3 instead?

- Line 124: Should this not be days per degree centigrade rather than degrees centigrade per day?

- Line 143: "among different groups of organisms" would be more precise here.

- Line 166: "potential biological phenology-related ecological and climatic implications" Having five adjectives modify a single noun is rarely a good idea. Please simplify our split up to improve ease of reading.

- Line 193: Why "expected" and not "observed"?

- Line 204: I think the fact that plant growth activity is essentially NDVI data should be explained very early on in the main text as this is key for a good understanding of this study.

- Line 227: I also think the fact that "soil fauna" throughout the text refers to only the subset listed here should be mentioned earlier on. For example, nematodes, by far the most abundant and arguably functionally most important soil fauna, are not included in this study, if I understand correctly. This is not necessarily a criticism, as I realize one cannot study everything, but it should be very clear to the reader.

- Line 247: how can you construct total plant biomass if the shoot mass is based on 0.5 m² and the root mass on 98 cm²? Or were the shoot and root values extrapolated to a same surface area before summing?

- Line 262: The Kempson method is good for arthropods but not for enchytraeids and nematodes. See also a previous comment about this. I think you should make the readers very well aware that your data probably only represent a fraction of the actual soil fauna present.

- Line 265: Please explain briefly how body length and abundance were converted into biomass estimates.

- Line 273: I think "a random intercept per plot-month combination" or similar would be less confusing than calling it "nested". "Plots nested within sampling time" implies that different plots were sampled every sampling time, which I assume was not the case.

- Line 278: Again, I do not see the reasoning for changing to T-tests and Bonferroni corrections here, and think the same mixed models with Turkey tests would be more appropriate to analyze the soil temperature and water data.

- Line 280: I have a serious problem with these correlation analysis, as they seem to violate a Pearson's correlation assumption of data independence. The overall correlations (top panels in Fig. 3) definitely do, but also the correlations per warming treatment lump 24 data points, i.e. values from 6 months times 4 plots, into a single correlation, without taking into account the non-independence of data stemming from the same plot. The authors clearly understand this issue, because their mixed models to test climate and time effects on the three organisms groups use an appropriate random effects structure to account for this. You can only solve this problem by averaging the data used for the correlations per month across plots or per plot across months but then you only have 6 and 4 data values per correlation, respectively, which drains statistical power. Another option would be to build a mixed model with the same random effects structure as the other mixed models, and vary the independent and dependent variable. For example, yes effect of NDVI on microbial respiration, and vice versa.

- Line 394: Table 1 lists "F-value" twice for plant growth activity and soil microbial respiration activity. I assume the second instance should be "P-value" here. I also suggest to have a few more significant digits for the P-values. Especially for the 0.05 for the climate by time interaction for soil

fauna it is not clear whether the actual value is higher or lower than 0.05 before rounding.

- Line 396: "all log-scaled". I assume this was done to improve normality of model residuals. If so, please add how normality of residuals was tested and what the test results were.
- Line 401: Perhaps a matter of taste, but I think the left column of panels shows all there is to show. I do not think the panels where data are collapsed per warming treatment or per sampling time add much information.
- Line 471: What is the rationale for using T-tests with Bonferroni corrections here? These temperature were collected from the same plots, so I would suggest you use the same, appropriate mixed model as you did for the soil organisms' activities, taking the plot random effect into account and apply Tukey tests to correct for post hoc multiple comparisons. I prefer FDR over Turkey corrections for multiple testing, and Kenward-Roger over Sattthertwaite approximation of df, but that is a detail.
- Line 484: The caption to Supp. Fig. 5 contains a lot of repetition that could be collapsed for easier reading.
- Line 156: Whether three years is immediate is relative. Perhaps "rather fast" responses would be more nuanced here.

Reviewer #2 (Remarks to the Author):

This manuscript evaluates changes in phenology in soil systems in response to warming. The authors present a unique experiment outlining a phonological disconnect between plants, microbes, and soil fauna under a warming regime. Their results show that phonological markers like plant growth, microbial respiration, and faunal activity increased under warming, but these changes were asynchronous. The authors present a well-designed study, to my understanding represents a pilot to future in depth studies. One thing lacking from the experimental design, in my opinion, was the absence of changes in precipitation patterns. Warming will inevitably impact precipitation cycles, so it would be stronger if some form of manipulation was added to test this. This is referenced at L51-52. I think the choice to not include increased or decreased precipitation events should be explained in the text. The other concern I have with the paper is the frequent use of the term "positive warming effects". I am not clear what this means in the context of the paper. My initial interpretation is that this refers to an increase in activity or a positive feedback loop. But it appears sometimes to mean positive i.e. good impacts of warming as a potential offset for other environmental changes. I would suggest rephrasing this throughout the paper with more precise language (i.e. L52, 86).

Minor corrections:

L46 – space between last letter and reference

L71 – please define NVDI here

L90 – as before, what does this mean under expected changes in precipitation rate?

L96-101 – what are the potential mechanisms behind this?

L185 – was there variation between plots? Or were they all relatively similar in regards to physicochemical properties

L207 – please define RGB.

Reviewer #3 (Remarks to the Author):

Review of the manuscript titled "Climate warming causes mismatches in plant-microbe-fauna phenology", by Yin et al.

My review is focused on camera-NDVI retrieval, as specifically asked by the editor.

Camera-derived NDVI obtained from NIR-enabled digital camera represents a valuable complement to the traditional greenness index or green chromatic coordinate, derived from visible RGB channels.

The method for the extraction of camera-NDVI from a stack of images consists of 5 steps. 1) RGB and RGB + NIR images must be paired, DN extracted from both, 2) DN must be corrected for exposure and 3) pseudo-NDVI can be calculated. 4) Because digital numbers are not a measure of radiances, a scaling factor is needed to scale NDVI to "normal" values in order to be comparable with other NDVI sources, e.g. satellite derived NDVI or field spectral sensors, 5) a severe filtering process must be applied to obtain a clean NDVI time series as the one depicted in fig S3.

This rather long introduction serves to highlight how the methodology presented in the manuscript by Yin et al. is clearly insufficient to understand how the authors proceeded in the NDVI retrieval from camera images.

The five steps illustrated above must be undertaken, or a valid reason for an alternative strategy must be clearly explained in the methods, or in the supplementary material.

In addition to that, I am also really surprised about how perfect is the NDVI signal retrieved from the camera.

I have processed some 74 site-years of camera NDVI data (many of them from grasslands) and never seen such clean time series. The authors state that "The dots represent the daily maximum NDVI values for each plot" (caption of supplementary figure S3). I would be curious to see the raw sub-daily time series of NDVI from which these daily maximum values are retrieved. The data points in the non growing season are almost perfect on a baseline, which is really astonishing to me.

In summary, the methods used to retrieve, pre-process and finally get the time series shown in figure s3 must be explained and documented with plots and statistics in order to convince the reader of the soundness of the procedure.

Responses (in blue font) to the comments by the reviewers

REVIEWER COMMENTS

Reviewer #1 (Remarks to the Author):

The authors used a soil warming experiment in an alpine system to assess how different groups of organisms (plants, soil microbes, and soil fauna) respond to warming not only in terms of their activity and biomass but also the timing of this activity. The authors find that while plants and soil microbes show an increased peak of activity under warming, this happens at the same time as in the control plots, namely in the middle of the summer season. Soil fauna, in contrast show a shift in the timing of the activity peak from mid season to early and late season, without the two new peaks differing in magnitude from that in the control plots. They conclude that this phenological mismatch under warming could have important consequences for ecosystem functioning in such alpine systems. I agree with the authors that this is an important topic and experiments such as the one presented here are the way forward to increase our limited understanding of the effects of climate warming on soil organisms and processes. In general, I like the straightforward setup of the experiment and the uncomplicated way in which the results are presented. In general the manuscript is well prepared. However, I have a few major and some minor comments that preclude me to recommend the manuscript in its current form for publication in Nature Communications. If the authors can address these concerns, I think the manuscript can represent an important contribution to the development of this understudied field.

Major comments:

- Would the fact that you measured root biomass to a depth of 30 cm, soil microbial activity from a depth of 65 cm and soil fauna activity only to a depth of 10 cm have influenced the results, and if so, how?

Response: Good point. We do, however, not think that this had much or any effect on our results. We measured these variables at ecologically-relevant soil depths. Root biomass was measured to a depth of 30 cm because in our alpine meadow of the Qinghai-Tibetan Plateau, 96% of root biomass is typically found in the upper 30 cm (Yang et al., 2009); hence, this is where most or all of plant growth activity takes place.

The soil is clay-loam, and its average depth is 65 cm; soil microbial activity was therefore measured to a depth of 65 cm in line with previous studies (e.g., Li et al., 2020; Wang et al., 2020, 2014) showing that also at deeper soil depth there could be some, although not much, microbial activity.

Soil fauna activity was measured to a depth of 10 cm using standard bait lamina strips. The bait substrate is primarily consumed by Collembola, Acari, enchytraeids, millipedes, and earthworms. In line with previous studies (e.g., Thakur et al., 2018; Siebert et al., 2019), these strips were employed in the upper soil layer because this is where these soil animals are most active. Thus, we think that we have measured the presence of roots and the activities of the soil biota to the relevant soil depths. To address this point, we have added some more explanation in the Methods to clarify why we think our measurement of root biomass (**Lines 283-288**), soil microbial activity (**Lines 248-249**), and soil faunal activity (**Lines 257**) were accurate and appropriate in addressing our research questions.

References

- Li, F. et al. Warming alters surface soil organic matter composition despite unchanged carbon stocks in a Tibetan permafrost ecosystem. *Funct. Ecol.* 34, 911-922 (2020).
- Siebert, J. et al. Extensive grassland-use sustains high levels of soil biological activity, but does not alleviate detrimental climate change effects. *Adv. Ecol. Res.* 60, 25–58 (2019).
- Thakur, M. P. et al. Reduced feeding activity of soil detritivores under warmer and drier conditions. *Nat. Clim. Chang.* 8, 75–78 (2018).
- Wang, C., Ren, F., Zhou, X., et al. Variations in the nitrogen saturation threshold of soil respiration in grassland ecosystems. *Biogeochemistry* 148, 311-324 (2020).
- Wang, Y. et al. Non-growing-season soil respiration is controlled by freezing and thawing processes in the summer monsoon-dominated Tibetan alpine grassland. *Global Biogeochem. Cycles* 28, 1081-1095 (2014).
- Yang, Y., Fang, J., Ji, C., & Han, W. Above-and belowground biomass allocation in Tibetan grasslands. *J. Veg. Sci.* 20, 177-184 (2009).

- The soil fauna activity was estimated with bait-lamina strips and their abundance with Kempson extractions. Both methods are quite selective, mainly targeting soil (micro)arthropods, but ignoring abundant groups such as nematodes, rotifers, tardigrades etc., many of which feed on microbes and therefore would be relevant to this study. This fact should at least be highlighted and its possible implications discussed.

Responses: Thank you very much for this comment. The bait substrate (from bait-lamina strips) is primarily consumed by soil invertebrate decomposers, such as earthworms, isopods, millipedes, enchytraeids, Collembola, and oribatid mites. However, we acknowledge that the Kempson method is limited to extractions of (micro)arthropods. Nevertheless, both methods are designed to assess abundance and activity of a wide range of soil fauna whilst a number of specific methods would be necessary to investigate the remaining groups. In the revised manuscript, we have addressed this point, highlighting that other taxonomic groups (beyond microarthropods) are also very important for ecological processes, such as decomposition rate and energy flux. For instance, predatory nematodes feed on some invertebrate detritivores, regulating the structure of soil food webs. Please see the **Lines 302-304**.

- Whereas the linear mixed models seem adequate, the correlation analyses do not. They seem to have been performed on data from across plots and months, essentially violating the assumption of data independence. Furthermore, all data for the soil organisms groups were log-scaled, presumably to improve normality of model residuals, but this is not explained nor are results of normality tests presented. See detailed comments about statistics below.

Responses: Thank you very much for your helpful suggestions on data analysis. Accordingly, we have re-analysed our data as follows:

- (i) Linear mixed models were used with Kenward-Roger's approximation of *df*, and FDR corrections for multiple testing;
- (ii) Correlation analyses were re-run by building a mixed model (`lmm1 <-lmer (microbial_activity ~ faunal_activity + (1|Block) + (1|Month), data)`) with the same random effects structure as the other mixed models.

In our revised manuscript, we have explained and described how we did the tests for the normality of residuals and homogeneity of variances, as well as why we log-transformed the data. Specifically, prior to analysis, all data were tested for normality of residuals using the Shapiro-Wilk test, and for homogeneity of variances using the `leveneTest()` function in the R *'car'* package. If the data did not conform to normality (i.e., $P < 0.05$ means the distribution of data is significantly different from the normal distribution), they were log-transformed ($\log(x + 1)$) to improve the fit of models. These analyses confirmed our initial findings and conclusions. Please, see Lines **308-312**. Please, see also our responses to your detailed comments below.

- Line 24: Suggest rephrasing as “many plants and aboveground animals” as otherwise the reader may think “aboveground” also applies to “plants”, which are both above- and belowground.

Responses: Nice suggestion! We have changed it to “many plants and aboveground animals”. Please, see **Line 24**.

- Line 33: Suggest removing “the activities”

Responses: Done.

- Line 35: Suggest rephrasing as “warming advanced soil fauna activity in the spring and delayed it in the autumn”

Responses: Thank you so much for helping us rephrase this sentence. We have made changes in the text as you suggested. Please, see Lines **35-36**.

- Line 39. The word “mismatch” is perhaps not ideal here, as from the previous sentence I gather that peak activity still occurs at the same time for all three tested groups of organisms.

Responses: We agree. We accordingly have replaced it with a more accurate word: asynchrony. Please, see **Line 38**.

- Line 46: Suggest replacing “altitudes” with “elevations”

Responses: Agreed. We have replaced it as you suggested.

- Line 48: “biological activities are expected to increase with phenological shifts” is slightly ambiguous. Do you mean that as phenological shifts increase, biological activities are expected to increase in tandem, or just that biological activities are expected to increase anyhow, accompanied by phenological shifts?

Responses: We apologize for the inaccuracy of this sentence. Yes, here we mean just that biological activities are expected to increase. We have rephrased this sentence as “Warming, however, can alleviate the thermal constraints, thus biological activities are expected to increase under predicted future warming scenarios”. Please, see Lines **48-49**.

- Line 51: Why soil moisture and not moisture in general? This paragraph is still talking about both above- and belowground organisms.

Responses: Correct! It should be “water” instead of “soil water” here. We have corrected this. Please, see **Line 50**.

- Line. 54: This is a bold claim about “organisms” in general with a single reference to a paper about

plant phenology in particular. I can come up with plenty of examples of phenology driven by day length and not temperature. Please tone down or be more precise.

Responses: We have rephrased the sentence as “Since temperature is one of the main drivers of the phenological cycle of organisms, on-going climate change may alter biological phenology with consequences for ecosystem functioning”. Please, see Lines **54-56**.

Additionally, we have added several more references (i.e., Parmesan and Yohe, 2003; Liu et al., 2022; Hegland et al., 2009; Vitasse et al., 2021;) to better support “organisms” in general (including plants, lichens, pollinators, reptiles, amphibians, birds, mammals, fishes, zooplankton, and marine invertebrates) in the rephrased sentence mentioned above. Please, see Lines **56-57**.

References

- Parmesan, C. & Yohe, G. A globally coherent fingerprint of climate change impacts across natural systems. *Nature* 421, 37–42 (2003).
- Liu, H. et al. Phenological mismatches between above- and belowground plant responses to climate warming. *Nat. Clim. Chang.* 12, 97–102 (2022).
- Hegland, S. J., Nielsen, A., Lázaro, A., Bjerknes, A. L. & Totland, Ø., How does climate warming affect plant-pollinator interactions? *Ecol. Lett.* 12, 184–195 (2009).
- Vitasse, Y. et al. Phenological and elevational shifts of plants, animals and fungi under climate change in the European Alps. *Biol. Rev.* 96, 1816–1835 (2021).

-Line 56. Same as previous comment, I think that “aboveground plants” makes no sense as plants are simultaneously above- and belowground (bar some epiphytes and other exceptions).

Responses: We agree and apologize for this mistake. We have rephrased the sentence as “In contrast to extensive recent work on the phenology of plants and aboveground animals in the context of climate warming, soil biota have rarely been included in phenological studies.” We have also added some more references (i.e., Wang et al., 2020; Samplonius et al., 2021; Youngflesh et al., 2021; Möhl et al., 2022) to support this statement. Please see the **Line 56**.

References

- Möhl, P., von Büren, R. S., & Hiltbrunner, E. Growth of alpine grassland will start and stop earlier under climate warming. *Nature communications*, 13(1), 1-10 (2022).
- Samplonius, J. M. et al. Strengthening the evidence base for temperature-mediated phenological asynchrony and its impacts. *Nat. Ecol. Evol.* 5, 155–164 (2021).
- Wang, H. et al. Alpine grassland plants grow earlier and faster but biomass remains unchanged over 35 years of climate change. *Ecol. Lett.* 23, 701–710 (2020).
- Youngflesh, C. et al. Migratory strategy drives species-level variation in bird sensitivity to vegetation green-up. *Nat. Ecol. Evol.* 5, 987–994 (2021).

- Line 63: “(i.e., plants, soil microbes, and soil fauna)” reads strange, as if this were the only way one could describe different groups of soil organisms. Perhaps “such as plants, soil microbes, and soil fauna” would be better.

Responses: We have changed it to “such as plants, soil microbes, and soil fauna” as you suggested. Please, see Lines **63-64**.

- Line 70: Suggest changing “whole-soil profile” to “whole-soil-profile”

Responses: Done.

- Line 71: “(i.e., NDVI)” suggests remote-sensing-based technology is equal to NDVI. NDVI is an index of plant productivity based on relationships among visible red and near infrared light reflected off of surfaces. Suggest changing to something like “and remote sensing measurements (e.g., NDVI)”

Responses: Yes, NDVI quantifies plant productivity by measuring the differences between near-infrared light reflected off of surfaces (which vegetation strongly reflects) and visible red light (which vegetation absorbs). In this study, we actually used PhenoCam-derived Normalized Difference Vegetation Index (NDVI), which complements the traditional greenness index or green chromatic coordinate, derived from visible RGB channels (see Reviewer 3).

For a more accurate description, we have corrected this sentence as “Based on in-situ observations and PhenoCam-derived Normalized Difference Vegetation Index (NDVI), the activity dynamics of plants growth, soil microbes, and soil fauna feeding were investigated to explore phenology shifts in the context of continuous warming.” Please, see Lines **71-73**.

- Line 77: This sentence has 52 words, no less. I suggest splitting into smaller, more digestible sentences.

Responses: Agreed. We have split this long sentence into two, as “More specifically, the increases in biological activities would be particularly pronounced in the early and late growing seasons, resulting in earlier spring phenology and delayed autumn phenology. Such a phenological shift pattern of alpine plant growth in response to natural climate warming has been revealed over the past few decades.” Please, see Lines **80-84**.

- Line 85: “soil fauna feeding” is too vague at this point without having read the methodology. Add briefly what you actually measured here.

Responses: We have adopted a more accurate term “soil invertebrate detritivore feeding” here based on published papers (Thakur et al., 2018; Siebert et al., 2019). Generally, earthworms, isopods, millipedes, enchytraeids, Collembola, and oribatid mites are considered to be the major soil invertebrate detritivores (Adl, 2003).

References

Adl, S. *The Ecology of Soil Decomposition* (CABI Publishing, Trowbridge, 2003).

Siebert, J. et al. Extensive grassland-use sustains high levels of soil biological activity, but does not alleviate detrimental climate change effects. *Adv. Ecol. Res.* 60, 25–58 (2019).

Thakur, M. P. et al. Reduced feeding activity of soil detritivores under warmer and drier conditions. *Nat. Clim. Chang.* 8, 75–78 (2018).

- Line 101: What is meant by “increased peaks”? Increased growth speed over a certain time interval?

Responses: It actually means “enhancement in maximum growth”. We have corrected this.

- Line 115: “off set” should be “offset”

Responses: Done.

- Line 119: The information that Fig. 2 is based on smoothing of the monthly mean values for ambient and warmed conditions is a bit limited. What was the smoothing procedure?

Responses: We have provided some more details in the figure legend: “The smoothed curves were plotted by stringing together the monthly average activities of these organisms under ambient and warmed climate conditions”. Please see the legend of Fig. 2 (**Lines 509-510**).

- Line 122: Should the reference to Supp. Fig. 4 here have been to Supp. Fig. 3 instead?

Responses: Yes! Thank you for spotting this mistake. We have corrected it accordingly.

- Line 124: Should this not be days per degree centigrade rather than degrees centigrade per day?

Responses: We apologize for this mistake, and have corrected the sentence to ‘Notably, climate warming has indeed advanced spring phenology of plant growth globally by an average of 4-6 days over the past few decades.’. Please, see Line **133**.

- Line 143: “among different groups of organisms” would be more precise here.

Responses: Done.

- Line 166: “potential biological phenology-related ecological and climatic implications” Having five adjectives modify a single noun is rarely a good idea. Please simplify or split up to improve ease of reading.

Responses: Thank you for your suggestion. We have changed it to “phenology-related implications”. Please, see Line **174**.

- Line 193: Why “expected” and not “observed”?

Responses: It should indeed be “observed”. We have corrected this. Please, see Line **204**.

- Line 204: I think the fact that plant growth activity is essentially NDVI data should be explained very early on in the main text as this is key for a good understanding of this study.

Responses: We agree. In the Introduction, we accordingly have explained this: “Based on PhenoCam-derived Normalized Difference Vegetation Index (NDVI) and *in-situ* investigations (combined with the dynamic chamber method and bait-lamina method), the activity dynamics of plant growth (represented by NDVI data), soil microbes, and soil fauna feeding (represented by the feeding of soil invertebrate detritivores, e.g., earthworms, isopods, millipedes, enchytraeids, Collembola, and oribatid mites) were investigated to explore phenology shifts in the context of continuous warming.”. Please see the **Lines 70-75**.

- Line 227: I also think the fact that “soil fauna” throughout the text refers to only the subset listed here should be mentioned earlier on. For example, nematodes, by far the most abundant and arguably functionally most important soil fauna, are not included in this study, if I understand correctly. This is not necessarily a criticism, as I realize one cannot study everything, but it should be very clear to the reader.

Responses: We now explain early in the manuscript that “soil fauna” refers to soil invertebrate detritivores, e.g., earthworms, isopods, millipedes, enchytraeids, Collembola, and oribatid mites. Please, see **Lines 73-75**. To note, these taxa are usually considered to be dominant soil invertebrate detritivores that occupy the most biomass in the soil (Adl, 2003), and they are the group of soil

invertebrates responsible for the substrate loss from bait-lamina strips in studies of soil animal feeding activity (e.g., Gongalsky et al., 2008; Birkhofer et al., 2011; Thakur et al., 2018; Siebert et al., 2019). Indeed, we must acknowledge that nematodes, as one of the most abundant and functionally important soil fauna, should also be considered in future such studies.

References

- Adl, S. The Ecology of Soil Decomposition (CABI Publishing, Trowbridge, 2003).
- Birkhofer, K. et al. Soil fauna feeding activity in temperate grassland soils increases with legume and grass species richness. *Soil Biol. Biochem.* 43, 2200–2207 (2011).
- Gongalsky, K. B., Persson, T. & Pokarzhevskii, A. D. Effects of soil temperature and moisture on the feeding activity of soil animals as determined by the bait-lamina test. *Appl. Soil Ecol.* 39, 84–90 (2008).
- Siebert, J. et al. Extensive grassland-use sustains high levels of soil biological activity, but does not alleviate detrimental climate change effects. *Adv. Ecol. Res.* 60, 25–58 (2019).
- Thakur, M. P. et al. Reduced feeding activity of soil detritivores under warmer and drier conditions. *Nat. Clim. Chang.* 8, 75–78 (2018).

- Line 247: how can you construct total plant biomass if the shoot mass is based on 0.5 m² and the root mass on 98 cm²? Or were the shoot and root values extrapolated to a same surface area before summing?

Responses: Shoot and root biomass were both expressed as gram per square meter. We have added more details to the description of our sampling design: ‘Mid-August 2021, we randomly chose two 50 cm × 50 cm quadrats in each plot to investigate aboveground productivity, and all plants of two diagonal 25 cm × 25 cm sub-quadrats in each quadrat were clipped to evaluate shoot biomass (g m⁻²). Additionally, five soil cores (Ø 5 cm, 30 cm depth) were taken in each plot to evaluate root biomass (g m⁻²). The sampling area of shoot and root in each plot was therefore 2500 cm² and 98 cm², respectively. Total plant biomass (g m⁻²) is equal to the sum of shoot biomass (g m⁻²) and root biomass (g m⁻²). Please, see **Lines 283-288**. This approach is in line with other studies evaluating above- and belowground plant biomass for the Tibetan alpine meadow (e.g., Liu et al., 2018; Yuan et al., 2020; Chen et al., 2021).

References

- Chen, Y. et al. Warming has a minor effect on surface soil organic carbon in alpine meadow ecosystems on the Qinghai–Tibetan Plateau. *Global Change Biology* 28, 1618–1629 (2021).
- Liu, H. et al. Shifting plant species composition in response to climate change stabilizes grassland primary production. *PNAS* 16, 4051–4056 (2018).
- Yuan, X. et al. Sensitivity of soil carbon dynamics to nitrogen and phosphorus enrichment in an alpine meadow. *Soil Biology & Biochemistry* 150, 107984 (2020).

- Line 262: The Kempson method is good for arthropods but not for enchytraeids and nematodes. See also a previous comment about this. I think you should make the readers very well aware that your data probably only represent a fraction of the actual soil fauna present.

Responses: You are right. In the revised version, we have mentioned that the Kempson method was used for the extractions of soil (micro)arthropods but not all faunal taxa. Please, see **Lines 302-304**.

- Line 265: Please explain briefly how body length and abundance were converted into biomass estimates.

Responses: We have briefly described how body length and abundance were converted into biomass: “Individual mean biomass (M , mg) was calculated using the formula $\text{Log } M = a + b \times \text{Log } L$, with L as the individual mean body length (μm), with Collembola: $a = 1.34$, $b = 1.99$; Oribatida: $a = 2.12$, $b = 2.71$; Prostigmata: $a = 2.12$, $b = 2.8$ (Mercer et al., 2001). Total biomass is equal to the number of individuals (abundance) multiplied by the individual mean biomass”. Please, see **Lines 306-309**.

Reference

Mercer R. D, Gabriel A. G. A., Barendse J., Marshall D. J. & Chown S.L. Invertebrate body sizes from Marion Island. *Antarctic Science* 13: 135–143 (2001).

- Line 273: I think “a random intercept per plot-month combination” or similar would be less confusing than calling it “nested”. “Plots nested within sampling time” implies that different plots were sampled every sampling time, which I assume was not the case.

Responses: Thank you for this comment. We have revised it as you suggested.

- Line 278: Again, I do not see the reasoning for changing to T-tests and Bonferroni corrections here, and think the same mixed models with Tukey tests would be more appropriate to analyze the soil temperature and water data.

Responses: We agree. Therefore, we have re-analysed our data (e.g., the data of biological activities, as well as soil temperature and moisture) using consistent linear mixed models with Kenward-Roger’s approximation of df , and FDR corrections for multiple testing. Please see the new **Table 1** and **Supplementary Fig. 2**. This reanalysis confirmed our previous results.

- Line 280: I have a serious problem with these correlation analysis, as they seem to violate a Pearson’s correlation assumption of data independence. The overall correlations (top panels in Fig. 3) definitely do, but also the correlations per warming treatment lump 24 data points, i.e. values from 6 months times 4 plots, into a single correlation, without taking into account the non-independence of data stemming from the same plot. The authors clearly understand this issue, because their mixed models to test climate and time effects on the three organisms groups use an appropriate random effects structure to account for this. You can only solve this problem by averaging the data used for the correlations per month across plots or per plot across months but then you only have 6 and 4 data values per correlation, respectively, which drains statistical power. Another option would be to build a mixed model with the same random effects structure as the other mixed models, and vary the independent and dependent variable. For example, yes effect of NDVI on microbial respiration, and vice versa.

Responses: Thank you very much for your guidance. We agree, and we have adopted the second option. Specifically, we have built a linear mixed model (like, `Lmm1 <- lmer (Plant_growth ~ microbial_activity + (1|Block) + (1|Month), data)`) with the same random effects structure as the other linear mixed models. We have also redone the graphs of the correlation analysis (please, see, **Fig. 3**).

Fig. 3 Correlations (R^2) between plant growth activity and soil microbial respiration activity **(a)**, between plant growth activity and soil fauna feeding activity **(b)**, between soil microbial respiration activity and soil fauna feeding activity **(c)**. Shown are the fitted regression lines and 95% confidence intervals in black for all plots and dates, in blue for ambient climate, and in red for warming climate; dots are represented by the corresponding colours of the two climates. Abbreviations: cR^2 = conditional R square; mR^2 = marginal R square.

- Line 394: Table 1 lists “F-value” twice for plant growth activity and soil microbial respiration activity. I assume the second instance should be “P-value” here. I also suggest to have a few more significant digits for the P-values. Especially for the 0.05 for the climate by time interaction for soil fauna it is not clear whether the actual value is higher or lower than 0.05 before rounding.

Responses: Yes, the second “F-value” should be “P-value”. We have corrected this. For the Climate \times Month interaction, the actual P-value (= 0.047985) is lower than 0.05. We have set this value to 0.048 in the revised version.

```
> anova(lmer)
Type III Analysis of Variance Table with Satterthwaite's method
      Sum Sq Mean Sq NumDF DenDF F value Pr(>F)
Climate      0.29197 0.291968      1      36 12.0149 0.001383 **
Month         0.53480 0.106960      5      36  4.4016 0.003140 **
Climate:Month 0.30433 0.060865      5      36  2.5047 0.047985 *
---
Signif. codes:  0 '***' 0.001 '**' 0.01 '*' 0.05 '.' 0.1 ' ' 1
```

- Line 396: “all log-scaled”. I assume this was done to improve normality of model residuals. If so, please add how normality of residuals was tested and what the test results were.

Responses: Yes. We have added more details: ‘Prior to analysis, all data were tested for normality of residuals using the Shapiro-Wilk test, and for homogeneity of variances using the leveneTest () function in the R ‘car’ package. If the data did not conform to normality and homogeneity of variances, they were log-transformed (log(x + 1)) to improve the fit of models’. Please see **Lines 312-316**.

- Line 401: Perhaps a matter of taste, but I think the left column of panels shows all there is to show. I do not think the panels where data are collapsed per warming treatment or per sampling time add much information.

Responses: With all respect, we would like to keep these right-most panels as they add some

interesting information. For instance, these results intuitively show that warming significantly increased plant growth in the early growing season (June and July) (**Fig. 1g**), and consistently increased soil microbial respiration from late spring (June) to early autumn (September) (**Fig. 1h**). By contrast, the responses of soil fauna feeding to warming varied across the seasons, with significant increases in their activity only in early spring (May) and late autumn (October) (**Fig. 1i**).

- Line 471: What is the rationale for using T-tests with Bonferroni corrections here? These temperature were collected from the same plots, so I would suggest you use the same, appropriate mixed model as you did for the soil organisms' activities, taking the plot random effect into account and apply Tukey tests to correct for post hoc multiple comparisons. I prefer FDR over Turkey corrections for multiple testing, and Kenward-Roger over Sattthertwaite approximation of df, but that is a detail.

Responses: Good point! We have re-analysed all data with the same mixed model. To evaluate the effects of warming on soil temperature and soil water content, we have therefore used the same linear mixed model with Kenward-Roger's method (significant effects followed by FDR corrections with $P < 0.05$) as we did for the soil organisms' activities, taking the plot random effect into account. Please see **Supplementary Fig. 2** in the revised version.

- Line 484: The caption to Supp. Fig. 5 contains a lot of repetition that could be collapsed for easier reading.

Responses: We have shortened the caption of **Supplementary Fig. 5** as 'Correlations (R) between soil abiotic factors (i.e., monthly mean temperature and water content at 5 cm soil depth) and plant growth activity (**a-b**), and soil microbial respiration activity (**c-d**), and soil fauna feeding activity (**e-f**).'

- Line 156: Whether three years is immediate is relative. Perhaps "rather fast" responses would be more nuanced here.

Responses: We have changed "immediate" to "rather fast".

Reviewer #2 (Remarks to the Author):

This manuscript evaluates changes in phenology in soil systems in response to warming. The authors present a unique experiment outlining a phenological disconnect between plants, microbes, and soil fauna under a warming regime. Their results show that phenological markers like plant growth, microbial respiration, and faunal activity increased under warming, but these changes were asynchronous. The authors present a well-designed study, to my understanding represents a pilot to future in depth studies. One thing lacking from the experimental design, in my opinion, was the absence of changes in precipitation patterns. Warming will inevitably impact precipitation cycles, so it would be stronger if some form of manipulation was added to test this. This is referenced at L51-52. I think the choice to not include increased or decreased precipitation events should be explained in the text.

The other concern I have with the paper is the frequent use of the term “positive warming effects”. I am not clear what this means in the context of the paper. My initial interpretation is that this refers to an increase in activity or a positive feedback loop. But it appears sometimes to mean positive i.e. good impacts of warming as a potential offset for other environmental changes. I would suggest rephrasing this throughout the paper with more precise language (i.e. L52, 86).

Responses: Thank you very much for your helpful comments and constructive suggestions, as well as your appreciation of our study. We indeed hope this study set the stage and break new ground for future in-depth studies of belowground phenology as you commented. Additionally, we have carefully addressed all your concerns below.

(1) Reply to your main concern 1

We agree with you that including precipitation patterns would have been of interest. But, that would be beyond the scope of this study. We have, however, explained this shortcoming in the text, as: ‘Note that: despite warming may inevitably affect precipitation patterns, our study did not include manipulation of precipitation due to inconsistent/unsure precipitation projections for this region (Kuang et al., 2016)’.

According to a review by Kuang et al. (2016), the Tibetan Plateau was overall getting warmer and potentially somewhat wetter during the last half century. Temperature significantly increased, and the overall warming rate ranged from 0.16 to 0.67°C per decade since the 1950s. The precipitation on the Tibetan Plateau slightly increased, but this increase was not as pronounced as that of temperature. Most studies have report that the increase in precipitation was not significant (Kuang et al., 2016). Furthermore, the pattern of precipitation changes on the Tibetan Plateau is far less uniform than that of temperature changes. Specifically, the temporal and spatial patterns of changes in precipitation are complicated, and the annual precipitation does not show any uniform increasing or decreasing trend across the Tibetan Plateau. Precipitation in some subregions could increase, while in some other subregions it may decrease. There is also not a uniform change in precipitation in different seasons. This factor was therefore not included in the study.

References

Mercer R. D., Gabriel A. G. A., Barendse J., Marshall D. J. & Chown S.L. Invertebrate body sizes from

Marion Island. *Antarctic Science* 13: 135–143 (2001).
Kuang X. & Jiu J. Review on climate change on the Tibetan Plateau during the last half century. *Journal of Geophysical Research: Atmospheres* 121: 3979–4007 (2016).

(2) Reply to your main **concern 2**

We apologize for the confusion. Accordingly, we have rephrased these sentences throughout the manuscript with more precise language. For example, we have (i) changed ‘Water scarcity during the dry summer months may become a limiting factor, consequently offsetting the positive warming effects’ to ‘Water scarcity during the dry summer months may be a limiting factor that offsets or diminishes the increases in biological activities caused by warming’; (ii) changed ‘warming may not always lead to positive effects in cold biomes, particularly in dry habitats or under dry conditions’ to ‘However, warming may not always increase biological activities in cold biomes, particularly in dry habitats or under dry conditions’.

Minor corrections:

L46 – space between last letter and reference

Responses: Done.

L71 – please define NVDI here

Responses: We have described it as ‘Normalized Difference Vegetation Index (NDVI)’. Please see **Line 71**.

L90 – as before, what does this mean under expected changes in precipitation rate?

Responses: Warming slightly reduced water content in the surface soil (0-10 cm) by about 6% across all plots and days, with only a significant reduction in the late growing season (September and October) (**Supplementary Fig. 2b**). This finding may suggest that warming influences soil moisture in a season-specific way.

L96-101 – what are the potential mechanisms behind this?

Responses: Earlier phenology of plant growth and their enhanced maximum growth during the alpine growing season do not necessarily mean an increased annual biomass production of plants. Warming may have led to increased water stress during the middle and/or late alpine growing season (Ernakovich et al., 2014). Consistent with this mechanism, we found a significant reduction in soil water content during the late growing season (i.e., in September and October, **Supplementary Fig. 2**), which was attributed to (1) the reduction induced by warming per se, and (2) greater consumption of available soil water by the warming-induced higher spring biomass (Wang et al., 2020). For example, in the same region, Wang et. al. (2020) found that the growth patterns of alpine grassland plants strongly responded to climate warming, and that the changes in growth patterns led to altered seasonal biomass production: spring production increased, summer production remained relatively constant, and autumn production decreased over time in this alpine grassland. Nevertheless, there is a lack of systematic change in annual biomass production. We have added some discussion on this topic to **Lines 105-110**.

References

Ernakovich, J. G. et al. Predicted responses of arctic and alpine ecosystems to altered seasonality under climate change. *Glob. Change Biol.* 20, 3256–3269 (2014).

Wang, H. et al. Alpine grassland plants grow earlier and faster but biomass remains unchanged over 35 years of climate change. *Ecol. Lett.* 23, 701–710 (2020).

L185 – was there variation between plots? Or were they all relatively similar in regards to physicochemical properties

Responses: There was small variation in soil physical and chemical properties between plots, but they were rather similar. We have added some more details to the text (**Lines 192-196**): ‘soil organic carbon (SOC): $72.9 \pm 17.6 \text{ g kg}^{-1}$, extractable organic carbon (EOC): $433.6 \pm 38.3 \text{ mg kg}^{-1}$, total nitrogen (TN): $7.2 \pm 1.6 \text{ g kg}^{-1}$, extractable organic nitrogen (EON): $62.7 \pm 6.8 \text{ mg kg}^{-1}$, inorganic nitrogen (IN): $7.8 \pm 0.9 \text{ mg kg}^{-1}$, ammonium nitrogen ($\text{NH}_4^+\text{-N}$): $4.2 \pm 0.5 \text{ mg kg}^{-1}$, nitrate nitrogen ($\text{NO}_3^-\text{-N}$): $3.6 \pm 0.8 \text{ mg kg}^{-1}$, and pH value: 7.6 ± 0.1 in the surface soil (0-10 cm) of experimental plots’.

L207 – please define RGB.

Responses: Done.

Reviewer #3 (Remarks to the Author):

Review of the manuscript titled "Climate warming causes mismatches in plant-microbe-fauna phenology", by Yin et al.

My review is focused on camera-NDVI retrieval, as specifically asked by the editor.

Camera-derived NDVI obtained from NIR-enabled digital camera represents a valuable complement to the traditional greenness index or green chromatic coordinate, derived from visible RGB channels.

The method for the extraction of camera-NDVI from a stack of images consists of 5 steps. 1) RGB and RGB + NIR images must be paired, DN extracted from both, 2) DN must be corrected for exposure and 3) pseudo-NDVI can be calculated. 4) Because digital numbers are not a measure of radiances, a scaling factor is needed to scale NDVI to "normal" values in order to be comparable with other NDVI sources, e.g. satellite derived NDVI or field spectral sensors, 5) a severe filtering process must be applied to obtain a clean NDVI time series as the one depicted in fig S3.

This rather long introduction serves to highlight how the methodology presented in the manuscript by Yin et al. is clearly insufficient to understand how the authors proceeded in the NDVI retrieval from camera images.

The five steps illustrated above must be undertaken, or a valid reason for an alternative strategy must be clearly explained in the methods, or in the supplementary material.

Responses: We really appreciate your helpful comments and constructive suggestions. In order to address your concerns, we have reanalyzed our data based on the steps you suggested here and some relevant literature.

Specifically, we extracted PhenoCam-derived NDVI from a stack of images including the following steps: (1) the PhenoCam (NetCam SC; StarDot Technologies, Buena Park, California, USA) obtained one pair of RGB and RGB+NIR images per shot (Petach et al., 2014). Digital Number (DN) values (R_{DN} , G_{DN} , B_{DN}) of red, green and blue channels were extracted from RGB images; DN values $[(RGB+NIR)_{DN}]$ of red, green, blue, and near infrared channels were extracted from RGB+NIR images. (2) The DN values extracted from RGB and RGB+NIR images were respectively calibrated to R_{DN_E} , G_{DN_E} , B_{DN_E} , and $(RGB+NIR)_{DN_E}$ according to the exposure value (Luo et al., 2018). (3) After calculating the calibration value of NIR_{DN_E} , phenological camera $NDVI_C$ was calculated: $NDVI_C = \frac{NIR_{DN_E} - R_{DN_E}}{NIR_{DN_E} + R_{DN_E}}$

(Liu et al., 2017). (4) In order to be comparable with field spectral sensors or satellite derived NDVI, we used a scaling factor to scale the DN-derived $NDVI_C$ to normal NDVI values (Filippa et al., 2018). The scaling factor was obtained using a linear regression between spectral reflectance-derived $NDVI_R$ (RapidSCAN CS-45; Holland Scientific, NE, USA) and DN-derived $NDVI_C$ for the same regions of interest in our studied alpine grassland. (5) Based on the daily dynamic of NDVI (**Supplementary Fig. 4**), the daily maximum NDVI value from 11:30 to 14:30 was selected for each day to calculate a mean daily NDVI. In winter, NDVI values sometimes were less than 0 due to snow cover (Brown et al., 2017), and we replaced the negative NDVI values with the 5% quantile of the remaining data (Filippa et al.,

2018). We also used the 95% quantile of the remaining data as the upper limit of winter NDVI. (6) The Savitzky-Golay filter with a formula: $f(t) = mn + (mx - mn) \cdot \left(\frac{1}{1 + e^{(-rep(t-sos) + \frac{1}{1 + e^{(rau(t-eos) - 1)}}} \right)$ in the R 'phenofit' package was used to filter and denoise NDVI sequences (Kong et al., 2022), and to fit the double logistic (DL) function, thereby obtaining smooth curves of NDVI time series (Beck et al., 2006). Please also see the detailed description in **Lines 216–244** in the revised version.

Supplementary Fig. 4 The PhenoCam shot eight times (at 7:00, 8:30, 10:00, 11:30, 13:00, 14:30, 16:00, 17:30) a day for each plot, and thus eight NDVI values were retrieved. Based on the NDVI sub-daily series, daily maximum NDVI values (from 11:30 to 14:30) were used to calculate a mean daily NDVI.

References

- Beck, P. S. A., Atzberger, C., Høgda, K. A., Johansen, B. & Skidmore, A. K. Improved monitoring of vegetation dynamics at very high latitudes: A new method using MODIS NDVI. *Remote Sensing of Environment* 100, 321–334 (2006).
- Brown, L. A., Dash, J., Ogutu, B. O. & Richardson, A. D. On the relationship between continuous measures of canopy greenness derived using near-surface remote sensing and satellite-derived vegetation products. *Agricultural and Forest Meteorology* 247, 280–292 (2017).
- Filippa, G. *et al.* NDVI derived from near-infrared-enabled digital cameras: Applicability across different plant functional types. *Agricultural and Forest Meteorology* 249, 275–285 (2018).
- Kong, D. *et al.* phenofit : An R package for extracting vegetation phenology from time series remote sensing. *Methods Ecol. Evol.* 13, 1508–1527 (2022).
- Liu, Y. *et al.* Using data from Landsat, MODIS, VIIRS and PhenoCams to monitor the phenology of California oak/grass savanna and open grassland across spatial scales. *Agricultural and Forest Meteorology* 237–238, 311–325 (2017).

Luo, Y. *et al.* Using Near-Infrared-Enabled Digital Repeat Photography to Track Structural and Physiological Phenology in Mediterranean Tree–Grass Ecosystems. *Remote Sensing* 10, 1293 (2018).

Petach, A. R., Toomey, M., Aubrecht, D. M. & Richardson, A. D. Monitoring vegetation phenology using an infrared-enabled security camera. *Agricultural and Forest Meteorology* 195–196, 143–151 (2014).

In addition to that, I am also really surprised about how perfect is the NDVI signal retrieved from the camera. I have processed some 74 site-years of camera NDVI data (many of them from grasslands) and never seen such clean time series.

The data points in the non growing season are almost perfect on a baseline, which is really astonishing to me.

Responses: The ‘perfect’ smooth curves of NDVI time series (**Supplementary Fig. 3b**) were not obtained from original NDVI_c of phenological camera; they were actually obtained by fitting the NDVI sequences (NDVI in winter was set as a constant in fitting) with the double logistic function (**Supplementary Fig. 3b**) using the Savitzky-Golay filter with a formula: $f(t) = mn + (mx - mn) \cdot \left(\frac{1}{1 + e^{(-rep(t-sos))}} + \frac{1}{1 + e^{(rau(t-eos))}} - 1 \right)$ in the R ‘phenofit’ package. Additionally, we have shown the curves of original NDVI_c time series of phenological camera in **Supplementary Fig. 3a** in this response letter.

Supplementary Fig. 3 NDVI_c time series of phenological camera (a), and NDVI time series with fitting a double logistic function (b) under ambient (in blue) and warmed (in red) conditions. The dots represent the daily maximum NDVI values for each plot. The blue and red curves are drawn by the daily NDVI means of ambient climate and warming climate, respectively. Abbreviations: EGS = early growing season (spring = May and June), MGS = middle growing season (summer = July and August), and LGS = late growing season (autumn = September and October).

The authors state that "The dots represent the daily maximum NDVI values for each plot" (caption of supplementary figure S3). I would be curious to see the raw sub-daily time series of NDVI from which these daily maximum values are retrieved.

Responses: In our study, the PhenoCam shot eight times (7:00, 8:30, 10:00, 11:30, 13:00, 14:30, 16:00, 17:30) a day for each plot, and therefore delivered eight corresponding NDVI values. Based on the NDVI sub-daily series (please, see the new added figure in the revised manuscript - **Supplementary Fig. 4**), we only used daily maximum NDVI values from 11:30 to 14:30 to gain daily mean maximum NDVI (Brown et al., 2017).

Supplementary Fig. 4 The PhenoCam shot eight times (at 7:00, 8:30, 10:00, 11:30, 13:00, 14:30, 16:00, 17:30) a day for each plot, and thus eight NDVI values were retrieved. Based on the NDVI sub-daily series, daily maximum NDVI values (from 11:30 to 14:30) were used to calculate a mean daily maximum NDVI.

References

Brown, L. A., Dash, J., Ogutu, B. O. & Richardson, A. D. On the relationship between continuous measures of canopy greenness derived using near-surface remote sensing and satellite-derived vegetation products. *Agricultural and Forest Meteorology* 247, 280–292 (2017).

In summary, the methods used to retrieve, pre-process and finally get the time series shown in figure s3 must be explained and documented with plots and statistics in order to convince the reader of the soundness of the procedure.

[redacted]

Dear [redacted]

Thank you so much again for your helpful comments and constructive suggestions. We have strictly reprocessed our data based on your guidance (the five steps for the extraction of camera-NDVI). The methods have been well explained in the new version as described above. We really hope you are satisfied with our revision.

Yours sincerely,
Biao (on the behalf of all co-authors)

Response from Reviewers, further round review:

Reviewer #1 (Remarks to the Author):

The authors did a commendable job revising this manuscript, which I believe has improved considerably as a result. I still have a few outstanding issues that I would like to see addressed before I can recommend this manuscript for publication:

- Line 308: "Prior to analysis, all data were tested for normality of residuals"

This sounds strange, as the sequence would be: do the analysis, store the residuals, test the residuals for normality. If not normally distributed, rerun analysis with log-transformed data, store new residuals, and test them for normality. In that sense I do not understand how you can test residuals before you even have done the analysis.

- Line 105. "(e.g., enhancement in maximum growth) of plant growth" sounds strange now ("growth ... of growth"). Why not write "... in the timing (e.g., earlier phenology) and increases in the maximum values of plant growth".

- Regarding the change to the legend of Fig.2, it now reads: "The smoothed curves were plotted by stringing together the monthly average activities of these organisms under ambient and warmed climate conditions". However, the means for each month are not just connected by straight lines, i.e., they are not barely "strung together". So I feel like the response still does not answer the original question: how was this smoothing achieved? It seems that some sort of smoothing function was applied.

- This new sentence confuses me (line 316): "In the LMMS, every two plots with different climate scenarios were randomly nested in each block, and block was served as a random effect." The original manuscript nowhere mentioned blocks. The way I understand it, you have 8 plots, 4 warming and 4 control plots, with repeated measurements over time for each plot. The logical model here would be time + climate + time x climate as fixed effects, as you describe. To account for the repeated measures per plot, a simple "plot" random effect would suffice. Unless you have multiple data points per plot per time point (in which case a time x plot random interaction would also be necessary), but it does not look that way. There was only one measurement of NDVI and soil respiration per plot per time. The bait lamina data were combined across 6 strips per plot per time. Root and shoot biomass were calculated in g per m² for the entire plot, each time. For the soil samples for soil microbial biomass (line 286) this is less clear. Perhaps indicate here how many samples were taken and if/how the data were pooled per plot. For soil fauna, 2 cores were taken per plot, but I assume that total biomass was calculated as one value per plot, based on all individuals summed from the 2 cores, at each time point. Please clarify. If indeed I understand correctly that the final data set for analysis had only one data value per month per plot for each variable, then a simple "plot" random effect would be enough I think. New Sup. Fig. 1 suggests you have now grouped plots per two into blocks. I think this is only meaningful if you know there is some gradient in the overall site that makes the two plots within block one more similar to each other than to plots in other blocks (e.g., a soil, moisture, or temperature gradient). If not, you are needlessly complicating things. If you do decide to stick with these blocks, the random effect would become "plot nested within block" (or a "block x climate" random interaction, which would achieve the exact same thing). A mere "block" effect, as you describe in your response ("lmm1 <- lmer (Plant_growth ~ microbial_activity + (1|Block) + (1|Month), data)") still does not take into account that the different measurements from different months from plot 1 are likely more intercorrelated than they are correlated with repeated measurements from plot 2.

Reviewer #3 (Remarks to the Author):

Review of the manuscript titled "Climate warming causes mismatches in plant-microbe-fauna phenology", by Yin et al.

My review is focused on camera-NDVI retrieval, as specifically asked by the editor.

The authors provide sufficient theoretical explanation of the processing steps for camera NDVI. However there are some points that must be yet clearly addressed.

1. Supplementary figure 3 in the manuscript must correspond to the one presented in the reviewer's response which is currently not the case.

2. Another supplementary figure is required showing:

a) Raw, uncorrected subdaily (NOTE: not just mid-day subset, but all available row data points) camera NDVI values from at least one plot

b) Raw midday values

c) The spectrometer derived NDVI data used to calibrate the scaling factors (lines 235-237) must be shown and details on how many points, how many plots, and so on must be shown. Details on instrumental setup are needed, too. How was this sensor installed? For how long, is it hand-held?

d) The scaling factor(s) must be reported in the main text, together with goodness of fit metrics.

3. Other things to be clarified.

"In winter, NDVI values sometimes were less than 0 due to snow cover (Brown et al., 2017), and we replaced the negative NDVI values with the 5% quantile of the remaining data (Filippa et al., 2018)."

There seems to be no below 0 values in NDVI trajectories, how many data points were actually replaced?

Keep separated the Savitzky-Golay filter by the double logistic formula you used to fit the data. State plainly that they consist of different procedures.

Responses (in red font) to the comments by the reviewers

REVIEWER COMMENTS

Reviewer #1 (Remarks to the Author):

The authors did a commendable job revising this manuscript, which I believe has improved considerably as a result. I still have a few outstanding issues that I would like to see addressed before I can recommend this manuscript for publication:

Response: Thank you very much for your valuable comments and constructive suggestions, as well as your appreciation of our revision work. In this round, we have carefully addressed all your concerns below. All line numbers refer to the clean version.

- Line 308: "Prior to analysis, all data were tested for normality of residuals"

This sounds strange, as the sequence would be: do the analysis, store the residuals, test the residuals for normality. If not normally distributed, rerun analysis with log-transformed data, store new residuals, and test them for normality. In that sense I do not understand how you can test residuals before you even have done the analysis.

Response: We apologize for this incorrect description. We have re-described statistical analyses, as below:

'Linear mixed effects models (LMMs, Type III ANOVA with Kenward-Roger's method) were used to analyze the fixed effects of climate (ambient vs. warming), sampling time (May, June, July, August, September, October), and their interactions on biological activities (i.e., plant growth, soil microbial respiration, soil fauna feeding) and other potential explanatory variables (i.e., soil fauna abundance and biomass, microbial biomass carbon, plant biomass) using the *lmer* () function in R '*lme4*' package⁶⁵. In the models, plot nested within block served as a random effect. The Shapiro-Wilk test and Levene's test were performed to test the normality of the model residuals and the homogeneity of variance, respectively. If not normally distributed, the data were log-transformed $\log(x+1)$ and the analyses were redone.'

Please see the line 322-330.

- Line 105. "(e.g., enhancement in maximum growth) of plant growth" sounds strange now ("growth ... of growth"). Why not write "... in the timing (e.g., earlier phenology) and increases in the maximum values of plant growth".

Response: Done! Please see the line 105-106. Thank you for restructuring and refining our sentence to make it read much better!

- Regarding the change to the legend of Fig.2, it now reads: "The smoothed curves were plotted by stringing together the monthly average activities of these organisms under ambient and warmed climate conditions". However, the means for each month are not just connected by straight lines, i.e., they are not barely "strung together". So I feel like the response still does not answer the original question: how was this smoothing achieved? It seems that some sort of smoothing function was applied.

Response: Fig. 2 is actually a concept-like diagram that allows readers to visualize our results (Fig. 1 g-i) easier and more intuitive. As we had only limited sampling data points (especially

for microbial respiration and fauna feeding, 6 times (monthly) during the growing season), we now simply connect the monthly mean values of these biological activities with straight lines to show their temporal dynamics. We had used a curved connection (just to look good) in the **previous Fig. 2**, and we now use the straight line-connection for the monthly mean values in the **new Fig. 2** to avoid misunderstanding, with additionally plotting the corresponding coloured sampling points ($n = 4$) for both ambient (in blue) and warming (in red) climates.

(new) Fig. 2 Phenology of activities of plants, soil microbes, and soil fauna during the growing season under ambient vs. warmed climate conditions. Warming increased the peak values of plant growth and soil microbial respiration activities in the middle of the growing season without much shifts in their phenology **(a, b)**. Warming advanced spring phenology and delayed autumn phenology of soil fauna feeding activity, with an unchanged activity peak during the growing season **(c)**. The monthly average activities of these organisms were connected by straight lines (with the corresponding coloured sampling points, $n = 4$) for both ambient and warming climates. Abbreviations: EGS = early growing season (spring = May and June), MGS = middle growing season (summer = July and August), and LGS = late growing

season (autumn = September and October).

- This new sentence confuses me (line 316): "In the LMMs, every two plots with different climate scenarios were randomly nested in each block, and block was served as a random effect." The original manuscript nowhere mentioned blocks. The way I understand it, you have 8 plots, 4 warming and 4 control plots, with repeated measurements over time for each plot. The logical model here would be time + climate + time x climate as fixed effects, as you describe. To account for the repeated measures per plot, a simple "plot" random effect would suffice. Unless you have multiple data points per plot per time point (in which case a time x plot random interaction would also be necessary), but it does not look that way. There was only one measurement of NDVI and soil respiration per plot per time. The bait lamina data were combined across 6 strips per plot per time. Root and shoot biomass were calculated in g per m² for the entire plot, each time. For the soil samples for soil microbial biomass (line 286) this is less clear. Perhaps indicate here how many samples were taken and if/how the data were pooled per plot. For soil fauna, 2 cores were taken per plot, but I assume that total biomass was calculated as one value per plot, based on all individuals summed from the 2 cores, at each time point. Please clarify. If indeed I understand correctly that the final data set for analysis had only one data value per month per plot for each variable, then a simple "plot" random effect would be enough I think. New Sup. Fig. 1 suggests you have now grouped plots per two into blocks. I think this is only meaningful if you know there is some gradient in the overall site that makes the two plots within block one more similar to each other than to plots in other blocks (e.g., a soil, moisture, or temperature gradient). If not, you are needlessly complicating things. If you do decide to stick with these blocks, the random effect would become "plot nested within block" (or a "block x climate" random interaction, which would achieve the exact same thing). A mere "block" effect, as you describe in your response ("lmm1 <- lmer (Plant_growth ~ microbial_activity + (1|Block) + (1|Month), data)") still does not take into account that the different measurements from different months from plot 1 are likely more intercorrelated than they are correlated with repeated measurements from plot 2.

Response: Thank you very much for your detailed description and guidance.

Yes! You understood correctly that the final data set for analysis had only one data value per month per plot for each variable. However, at the time our platform was built, there were indeed some natural gradients at the site (e.g. plant community composition, depth from soil surface to bedrock, microtopography) that made two plots within one block more similar to each other than to plots in other blocks. Moreover, each block (an ambient plot and a paired warming plot) was controlled by a separate computer program and datalogger system (as well as soil temperature and moisture sensors). Whether the warming cable for the warming plot in each block was switched on or off was determined independently by the temperature difference between the two plots (warming plot vs. ambient plot) within a block. In other words, the two paired plots within each of the four blocks were monitored (soil temperature and moisture) and controlled (turning on/off heater) independently. In this case, "plot nested within block" should be served as a random effect in the LMM models, and we have redone data analyses. Please see the **Line 327-330**, and new **Table 1**.

Additionally, we have clearly re-described the sampling for soil microbial biomass and soil fauna. Specifically, (1) five soil samples (Ø 3 cm, 10 cm depth) were randomly collected from each plot and evenly mixed (as a composite sample for this plot) to determine the content of

microbial biomass carbon by the chloroform fumigation-extraction method; (2) two soil cores (\varnothing 6 cm, 10 cm depth) were collected from each plot and placed together (as a composite sample for this plot) to extract soil fauna using a Kempson method. Please see the **Line 299-302, 310-312**.

Reviewer #3 (Remarks to the Author):

Review of the manuscript titled "Climate warming causes mismatches in plant-microbe-fauna phenology", by Yin et al.

My review is focused on camera-NDVI retrieval, as specifically asked by the editor.

The authors provide sufficient theoretical explanation of the processing steps for camera NDVI. However there are some points that must be yet clearly addressed.

1. Supplementary figure 3 in the manuscript must correspond to the one presented in the reviewer's response which is currently not the case.

Response: Done as you suggested! Please see the new **Supplementary Fig. 4**. Thank you for your very careful checking. All line numbers refer to the clean version.

2. Another supplementary figure is required showing:

a) Raw, uncorrected subdaily (NOTE: not just mid-day subset, but all available raw data points) camera NDVI values from at least one plot

b) Raw midday values

Response: Agreed! In the revised version, we have additionally supplemented a brief flowchart of NDVI data processing (**Supplementary Table 1**) that includes the uncorrected raw sub-daily and midday values. Please see the step 1 and 3 in **Supplementary Table 1**.

(new) Supplementary Table 1 A brief flowchart of NDVI data processing. Note: taking plot 3 and (paired) plot 4 (within block 2) to represent the ambient plots and the warming plots, respectively.

Description	Reference figures	
Step 1: Gaining phenological camera NDVI _c raw data.	 Plot 3 (Ambient)	 Plot 4 (Warming)
Step 2: Using a linear regression between Spectral reflectance NDVI _R and NDVI _c (Supplementary Fig. 6) to correct NDVI _c and gain normal NDVI values ⁵¹ .	 Plot 3 (Ambient)	 Plot 4 (Warming)
Step 3: Deleting the data points with NDVI ≤ 0 (usually due to snow cover in winter) and light deficiency-derived obvious outliers (like in cloudy days and early morning) ⁵² .	 Plot 3 (Ambient)	 Plot 4 (Warming)
Step 4: Extracting the daily maximum NDVI values (11:30 - 14:30) (Supplementary Fig. 7) from the remaining values.	 Plot 3 (Ambient)	 Plot 4 (Warming)
Step 5: Using the daily maximum NDVI values (11:30 - 14:30) to calculate mean daily NDVI values.	 Plot 3 (Ambient)	 Plot 4 (Warming)
Step 6: Using the 95% and 5% quantiles as the upper and lower limits of NDVI sequences to constrain the mean daily NDVI values ⁵¹ .	 Plot 3 (Ambient)	 Plot 4 (Warming)
Step 7: Using the Savitzky-Golay filter in the R 'phenofit' package to filter and denoise NDVI sequences ⁵³ .	 Plot 3 (Ambient)	 Plot 4 (Warming)
Step 8: Using the double logistic function to obtain the smooth seasonal dynamic curves of NDVI ⁵⁴ .	 Plot 3 (Ambient)	 Plot 4 (Warming)

c) The spectrometer derived NDVI data used to calibrate the scaling factors (lines 235-237) must be shown and details on how many points, how many plots, and so on must be shown. Details on instrumental setup are needed, too. How was this sensor installed? For how long, is it hand-held?

d) The scaling factor(s) must be reported in the main text, together with goodness of fit metrics.

Response: Agreed! This step has been further supplemented and re-described as low (line 235-242).

‘In order to be comparable with field spectral sensors or satellite derived NDVI, we scaled the digital number-derived $NDVI_C$ to normal NDVI values using a linear regression ($y = 0.9445x + 0.0471$, $n = 18$)⁵¹. This linear regression was established base on the data from simultaneous measurements of spectral reflectance-derived $NDVI_R$ (measured by Phenological camera, StarDot Technologies, Buena Park, California, USA) and digital number-derived $NDVI_C$ during the growing seasons of 2018 and 2019 (Supplementary Fig. 6) for the same regions of interest in our studied alpine grassland.’

Additionally, the spectrometer is handheld, and the phenological camera was installed in 2018 and has been in operation for five years (please see the legends of **Supplementary Fig. 6**).

(new) Supplementary Fig. 6 Linear regression of spectral reflectance-derived $NDVI_R$ (measured by RapidSCAN CS-45 handheld plant spectrometer; Holland Scientific, NE, USA) and digital number-derived $NDVI_C$ (measured by Phenological camera, StarDot Technologies, Buena Park, California, USA). The phenological camera was installed in 2018 and has been in operation for five years. A total of 18 plots was measured during the growing seasons of 2018 (11 yellow dots) and 2019 (7 green dots).

Supplementary Fig. 7 The PhenoCam shot and retrieved eight NDVI values (at 7:00, 8:30, 10:00, 11:30, 13:00, 14:30, 16:00, 17:30) within a day for each plot. Based on the NDVI sub-daily series, daily maximum NDVI values (from 11:30 to 14:30) were used to calculate a mean daily maximum NDVI.

3. Other things to be clarified.

"In winter, NDVI values sometimes were less than 0 due to snow cover (Brown et al., 2017), and we replaced the negative NDVI values with the 5% quantile of the remaining data (Filippa et al., 2018)."

There seems to be no below 0 values in NDVI trajectories, how many data points were actually replaced?

Response: Sorry for this confusion. Data points with $NDVI \leq 0$ had been deleted (not replaced) before calculating the daily mean NDVI values. For example, in plot 3 (ambient climate) and plot 4 (warming climate), the data points with $NDVI \leq 0$ accounted for 8.5% and 6.2% of the raw data, respectively. The 95% and 5% quantiles of all mean daily NDVI values were treated as the upper and lower limits of NDVI sequences to constrain mean daily NDVI values in a reasonable range. Please see the line **245-251** for a better understanding, we have provided a brief flowchart of NDVI data processing in **Supplementary Table 1** above.

Keep separated the Savitzky-Golay filter by the double logistic formula you used to fit the data. State plainly that they consist of different procedures.

Response: We have now re-described them to two separate steps, as 'The Savitzky-Golay filter in the R 'phenofit' package was used to filter and denoise NDVI sequences (Kong et al., 2022). The double logistic (DL) function was used to obtain the smooth seasonal dynamic curves of NDVI, and interpolate the missing values (Beck et al., 2006)'. Please see the line **251-253**.

We have also additionally provided a supplementary figure (**Fig. S8**) to intuitively show the raw, filtered, and fitted NDVI sequences under ambient and warming plots.

(new) Supplementary Fig. 8 Comparison for the raw, filtered, and fitted NDVI sequences under ambient (plot 3) and warming (plot 4) climates.

References

- Kong, D. et al. phenofit: An R package for extracting vegetation phenology from time series remote sensing. *Methods Ecol. Evol.* **13**, 1508–1527 (2022).
- Beck, P. S. A., Atzberger, C., Høgda, K. A., Johansen, B. & Skidmore, A. K. Improved monitoring of vegetation dynamics at very high latitudes: A new method using MODIS NDVI. *Remote Sens. Environ.* **100**, 321–334 (2006).

Response from Reviewers, final round review:

Reviewer #1 (Remarks to the Author):

The authors satisfactorily addressed all my previous comment, and I can now recommend this nice manuscript for publication. I have one last suggestion, though. In your rebuttal, you explain why you chose to group the plots per 2 into spatial blocks as follows:

"However, at the time our platform was built, there were indeed some natural gradients at the site (e.g. plant community composition, depth from soil surface to bedrock, microtopography) that made two plots within one block more similar to each other than to plots in other blocks. Moreover, each block (an ambient plot and a paired warming plot) was controlled by a separate computer program and datalogger system (as well as soil temperature and moisture sensors). Whether the warming cable for the warming plot in each block was switched on or off was determined independently by the temperature difference between the two plots (warming plot vs. ambient plot) within a block. In other words, the two paired plots within each of the four blocks were monitored (soil temperature and moisture) and controlled (turning on/off heater) independently. In this case, "plot nested within block" should be served as a random effect in the LMM models, and we have redone data analyses."

These are two very valid arguments for your choice of blocking: i) there are natural gradients in the field site, ii) the paired warmed and control plots were both steered by a single data logging system coupled to a warming system, which could cause data dependency (suppose, for example, that one of the systems malfunctions for a while). So why not just add this information to the methods section? Now the first reference to "block" is in the "statistical analysis" section. Other than that, the reader has to go to Supplementary Figure 1 to see what is meant by "block". You are making the reader work a bit hard here.

Reviewer #3 (Remarks to the Author):

Review of the manuscript titled "Climate warming causes mismatches in plant-microbe fauna phenology", by Yin et al.
My review is focused on camera-NDVI retrieval, as specifically asked by the editor.

The authors clarified all points raised in the processing of camera-NDVI. I think all these steps were needed in order to make the workflow clear, which was clearly not the case for the first version of the manuscript.

It is noteworthy however that the raw camera NDVI values fall within the range of spectral NDVI already before any correction. This must certainly have to do with some site/instrumental-specific feature. However, at present, as far as camera-NDVI is concerned, the manuscript and the data behind appear now sound. I suggest the publication of the manuscript.

Responses (in blue font) to the comments by the reviewers

Reviewer #1 (Remarks to the Author):

The authors satisfactorily addressed all my previous comment, and I can now recommend this nice manuscript for publication. I have one last suggestion, though. In your rebuttal, you explain why you chose to group the plots per 2 into spatial blocks as follows:

"However, at the time our platform was built, there were indeed some natural gradients at the site (e.g. plant community composition, depth from soil surface to bedrock, microtopography) that made two plots within one block more similar to each other than to plots in other blocks. Moreover, each block (an ambient plot and a paired warming plot) was controlled by a separate computer program and datalogger system (as well as soil temperature and moisture sensors). Whether the warming cable for the warming plot in each block was switched on or off was determined independently by the temperature difference between the two plots (warming plot vs. ambient plot) within a block. In other words, the two paired plots within each of the four blocks were monitored (soil temperature and moisture) and controlled (turning on/off heater) independently. In this case, "plot nested within block" should be served as a random effect in the LMM models, and we have redone data analyses."

These are two very valid arguments for your choice of blocking: i) there are natural gradients in the field site, ii) the paired warmed and control plots were both steered by a single data logging system coupled to a warming system, which could cause data dependency (suppose, for example, that one of the systems malfunctions for a while). So why not just add this information to the methods section? Now the first reference to "block" is in the "statistical analysis" section. Other than that, the reader has to go to Supplementary Figure 1 to see what is meant by "block". You are making the reader work a bit hard here.

Response: Thank you very much for your approval of our revision work. According to your further comments, we have added the description that you offered to explain why we chose to group the plots per 2 into spatial blocks in the Methods section. Please see the lines 202-205.

Reviewer #3 (Remarks to the Author):

Review of the manuscript titled "Climate warming causes mismatches in plant-microbe fauna phenology", by Yin et al.

My review is focused on camera-NDVI retrieval, as specifically asked by the editor.

The authors clarified all points raised in the processing of camera-NDVI. I think all these steps were needed in order to make the workflow clear, which was clearly not the case for the first version of the manuscript.

It is noteworthy however that the raw camera NDVI values fall within the range of spectral NDVI already before any correction. This must certainly have to do with some site/instrumental-specific feature. However, at present, as far as camera-NDVI is concerned, the manuscript and the data behind appear now sound. I suggest the publication of the manuscript.

Response: Thank you very much for your approval of our revision work. With your great help, our manuscript had been greatly improved in terms of NDVI data processing and description in the previous rounds of revisions.